# SPOP targets the immune transcription factor IRF1 for proteasomal degradation

Irene Schwartz[1,2†], Milica Vunjak[1,2†], Valentina Budroni[1,2†], Adriana Cantoran García[1], Marialaura Mastrovito[1], Adrian Soderholm[1,2], Matthias Hinterndorfer[2,3], Melanie de Almeida[2,3], Kathrin Hacker[1], Jingkui Wang[3], Kimon Froussios[3], Julian Jude[3], Thomas Decker[1], Johannes Zuber[3,4], Gijs A Versteeg[1]*

[1]Department of Microbiology, Immunobiology and Genetics, Max Perutz Labs, University of Vienna, Vienna, Austria; [2]Vienna BioCenter PhD Program, Doctoral School of the University of Vienna and Medical University of Vienna, Vienna Biocenter, Vienna, Austria; [3]Research Institute of Molecular Pathology, Vienna Biocenter, Vienna, Austria; [4]Medical University of Vienna, Vienna BioCenter, Vienna, Austria

*For correspondence:
gijs.versteeg@univie.ac.at

†These authors contributed equally to this work

## Abstract

Adaptation of the functional proteome is essential to counter pathogens during infection, yet precisely timed degradation of these response proteins after pathogen clearance is likewise key to preventing autoimmunity. Interferon regulatory factor 1 (IRF1) plays an essential role as a transcription factor in driving the expression of immune response genes during infection. The striking difference in functional output with other IRFs is that IRF1 also drives the expression of various cell cycle inhibiting factors, making it an important tumor suppressor. Thus, it is critical to regulate the abundance of IRF1 to achieve a 'Goldilocks' zone in which there is sufficient IRF1 to prevent tumorigenesis, yet not too much which could drive excessive immune activation. Using genetic screening, we identified the E3 ligase receptor speckle type BTB/POZ protein (SPOP) to mediate IRF1 proteasomal turnover in human and mouse cells. We identified S/T-rich degrons in IRF1 required for its SPOP MATH domain-dependent turnover. In the absence of SPOP, elevated IRF1 protein levels functionally increased IRF1-dependent cellular responses, underpinning the biological significance of SPOP in curtailing IRF1 protein abundance.

## Editor's evaluation

IRF1 is a key transcription factor with important roles during pathogenic infection and in some cancers. The regulation of IRF1 protein abundance is critical, but the underlying mechanisms are not known. Using a CRISPR-based knockout screen, the authors identify SPOP as an E3 ligase that binds IRF1 and enforces its degradation via the proteasome. The study represents an important advance backed by solid evidence that may have implications in immunology, virology, and cancer fields.

## Introduction

Adaptation of the functional proteome is essential to counter pathogens during infection, yet precisely timed degradation of these response proteins after pathogen clearance is likewise key to preventing autoimmunity and hyperinflammation (*Crow and Stetson, 2022*).

Recognition of pathogen-associated molecular patterns and subsequent secretion of cytokines known as interferons (IFNs) are particularly critical for conferring an anti-pathogen response (*Takeuchi and Akira, 2010*). Type I interferons (IFNα/β) can be secreted by most cells upon infection, whereas

type II IFNs (IFNγ) are secreted by specific cell types such as natural killer cells and T helper 1 cells. Binding of these cytokines to their receptors initiates cell signaling driving transcription of hundreds of response genes, which collectively mediate the anti-pathogen response (*Schoggins, 2019*).

Interferon regulatory factors (IRFs) play essential roles as transcription factors in the IFN system. Some IRFs, such as IRF3, are constitutively expressed. Their activity is predominantly regulated through phosphorylation, and they are essential for induction of type I IFNs (*Tamura et al., 2008*). In contrast, some other IRFs, including IRF1, are transcriptionally induced during infection (*Feng et al., 2021*).

IRF1 is a key family member for the induction of the innate immune response (*Forero et al., 2019*; *Kimura et al., 1994*; *Langlais et al., 2016*). The importance to precisely control IRF1 levels and its functional output is underpinned by the fact that aberrant IRF1 expression is linked to inflammatory syndromes, such as Behcet's disease, arthritis, and systemic lupus erythematosus (*Donn et al., 2001*; *Govind et al., 2014*; *Lee et al., 2007*; *Uddin et al., 2011*; *Zhang et al., 2015*). The striking difference in functional output with other IRFs is that IRF1 also drives the expression of various cell cycle inhibiting factors (*Armstrong et al., 2012*; *Tanaka et al., 1996*), making it an important tumor suppressor (*Nozawa et al., 1999*; *Xie et al., 2003*). Thus, it is critical to regulate the abundance of IRF1 to achieve 'Goldilocks' levels at which there is sufficient IRF1 to prevent tumorigenesis, yet not too much which could cause excessive immune activation.

In addition to its transcriptional regulation, the other main means of IRF1 regulation is at a protein level through rapid degradation by the proteasome (*Nakagawa and Yokosawa, 2000*; *Pion et al., 2009*), which seems to in turn be critical to limit transcriptional output and reduce the IRF1 protein pool once IRF1 transcription is shut off during immune resolution. In line with this notion, proteasome inhibitors have been shown to block IRF1 degradation (*Attard et al., 2005*). Several factors have thus far been reported to mediate poly-ubiquitination of IRF1, and direct its proteasomal degradation.

The importance of these individual regulators for IRF1 turnover is to a degree determined by their cellular context (*Liu et al., 2021*; *Remoli et al., 2020*). This draws an interesting parallel to other transcription factors which can be ubiquitinated by various different E3 ligase complexes, depending on cellular context. cMYC is a prime example of such a cell context-regulated transcription factor. A Cullin E3 ligase complex using the FBXW7 substrate receptor module (SCF$^{FBXW7}$) has been reported as a key factor in cMYC degradative ubiquitination in many cell types (*Welcker et al., 2004*; *Yada et al., 2004*). In contrast, various other E3s, including TRIM32, CHIP, and several other Cullin F-box complexes, have been shown to be important for cMYC degradation, yet in more cell-specific contexts (*Sun et al., 2021*). Identification and characterization of such E3 ligases is critical for ultimately understanding how their action controls transcriptional output in various cell types, within the context of whole organisms.

As for other transcription factors, turnover mechanisms can be different for distinct cellular IRF1 protein pools, dependent on their localization and functional engagement in transcription. As such, understanding which functional IRF1 domains determine its turnover is important. IRF1 has an N-terminal DNA binding domain, followed by a nuclear localization signal (NLS), IRF1 transactivation IRF-association domain (IAD2), and a C-terminal intrinsically disordered enhancer region (IDR). This C-terminal IDR is an important determinant of IRF1 turnover (*Nakagawa and Yokosawa, 2000*). On the one hand, the IDR is stabilized by the chaperones HSP70 and HSP90 (*Landré et al., 2013*; *Nakagawa and Yokosawa, 2000*; *Narayan et al., 2009*; *Narayan et al., 2015*; *Narayan et al., 2011a*; *Narayan et al., 2011b*), whereas it is targeted for degradative ubiquitination by the protein quality control E3 ligase CHIP (*Narayan et al., 2015*; *Narayan et al., 2011b*). This has suggested that cells may, at least in part, turn over IRF1 co- or post-translationally using their general protein quality control pathways.

In addition, the N-terminal DNA binding domain has been reported to determine IRF1 stability (*Landré et al., 2013*). Consistent with a model in which IRF1 is ubiquitinated by the E3 ligase MDM2 in an unbound state, DNA binding competes with E3 ligase interaction, thereby stabilizing IRF1 in a functionally engaged state (*Landré et al., 2013*).

Lastly, the IAD2 domain has been implicated in driving IRF1 ubiquitination and degradation (*Garvin et al., 2019*; *Pion et al., 2009*). In part, this has been attributed to generation of a GSK3β-dependent phospho-degron on DNA-bound IRF1, and subsequent recognition by the SCF$^{FBXW7}$ E3 ligase complex, which drives its degradation.

In this study, we set out to identify novel factors that control IRF1 abundance in an unbiased manner. Genome-wide genetic screens were performed to identify novel IRF1-specific E3 ligases.

This identified the Cullin ubiquitin E3 ligase receptor SPOP to functionally mediate IRF1 degradation. Follow-up analyses identified four degrons in, or near the IRF1 IAD2 enhancer domain responsible for SPOP binding, and subsequent IRF1 ubiquitination and degradation. Consequently, loss of *SPOP* increased transcriptional output of IRF1-stimulated genes.

## Results

### Establishment of an IRF1 protein abundance reporter cell line for genetic screening

The IRF1 protein is ubiquitinated and turned over by the proteasome. However, the cellular E3 ligases contributing to targeting IRF1 for degradation have remained poorly understood. The overarching goal of this study was to identify and characterize novel regulators of IRF1 protein abundance. For the identification of IRF1 stability regulators, we characterized IRF1 degradation properties in the human RKO colon carcinoma cell line, as it is highly amenable for genetic screening.

IRF1 is transcriptionally induced upon type I or type II IFN stimulation in a wide variety of cells. Likewise, stimulation of RKO cells with IFNγ strongly induced endogenous IRF1 protein levels (*Figure 1A*; 11.2-fold). Subsequent treatment with the translation inhibitor cycloheximide (CHX) rapidly chased out the induced IRF1 protein (*Figure 1A*). In the absence of de novo protein synthesis, IRF1 levels returned to pre-stimulation protein levels within 4 hr, indicating that it is highly unstable. Simultaneous inhibition of lysosomal degradation by Bafilomycin A (Baf) did not affect endogenous IRF1 stability and half-life (*Figure 1—figure supplement 1A–B*), indicating that IRF1 is predominantly turned over by the proteasome. In line with this finding, endogenous IRF1 protein levels measured by intracellular staining were also induced by IFNγ and depleted upon translation inhibition (*Figure 1B*; left panel), whereas inhibition of proteasomal degradation by MG132 increased IRF1 levels (*Figure 1B*; right panel). Together, these results indicate that IRF1 is induced by IFNγ and degraded by the proteasome in RKO cells, as has been described for other cell types, making it a suitable model for identification of IRF1 stability regulators.

Endogenous IRF1 protein levels are influenced by transcriptional, translational, and post-translational effects (*Feng et al., 2017*). Therefore, a strategy was devised to uncouple IRF1 protein abundance from its endogenous transcriptional program, thereby aiming to maximize the identification of regulators of protein degradation. To this end, a monoclonal cell line harboring Dox-inducible Cas9 was established, and a dual-color reporter construct under the control of a constitutively active promoter (*Figure 1C*), consisting of an mCherry-IRF1 cassette separated from BFP by a P2A ribosomal skip site. Although both proteins are translated from a single transcript in equimolar quantities, the unstable mCherry-IRF1 was predicted to accumulate at lower steady-state levels than the stable BFP internal control.

We reasoned that it would be important to screen in parallel with an unrelated protein that is degraded in a proteasome-dependent manner, in order to distinguish between general players in the ubiquitin proteasome system and factors specifically regulating IRF1 abundance. cMYC is constantly turned over by the proteasome with a 20–30 min half-life (*Gregory and Hann, 2000*; *Ramsay et al., 1986*). Therefore, a comparable cell line expressing mCherry-cMYC was established and characterized alongside the mCherry-IRF1 cell line.

To investigate whether the mCherry-tagged IRF1 reporter had comparable degradation characteristics as its endogenous counterpart, cells were subjected to a CHX chase to determine protein stability. In line with endogenous IRF1, mCherry-IRF1 levels rapidly diminished within 4 hr (*Figure 1D*). IRF1 degradation was prevented by simultaneous inhibition of the proteasome (*Figure 1D*), whereas inhibition of lysosomal degradation by Baf did neither stabilize endogenous IRF1 nor mCherry-IRF1 (*Figure 1E*). Similarly, mCherry-cMYC and endogenous cMYC were rapidly lost upon CHX treatment, which was prevented by simultaneous proteasome inhibition (*Figure 1—figure supplement 1C–D*). This indicated that mCherry-IRF1, mCherry-cMYC, and their endogenous equivalents are all predominantly degraded by the proteasome.

Next, we set out to determine whether our fluorescent reporters were compatible with fluorescence-activated cell sorting (FACS)-based genetic screening. In particular, we assessed by flow cytometry whether inhibition of protein synthesis or proteasomal degradation would specifically affect the mCherry-tagged unstable proteins, without affecting their stable BFP internal controls. Consistent

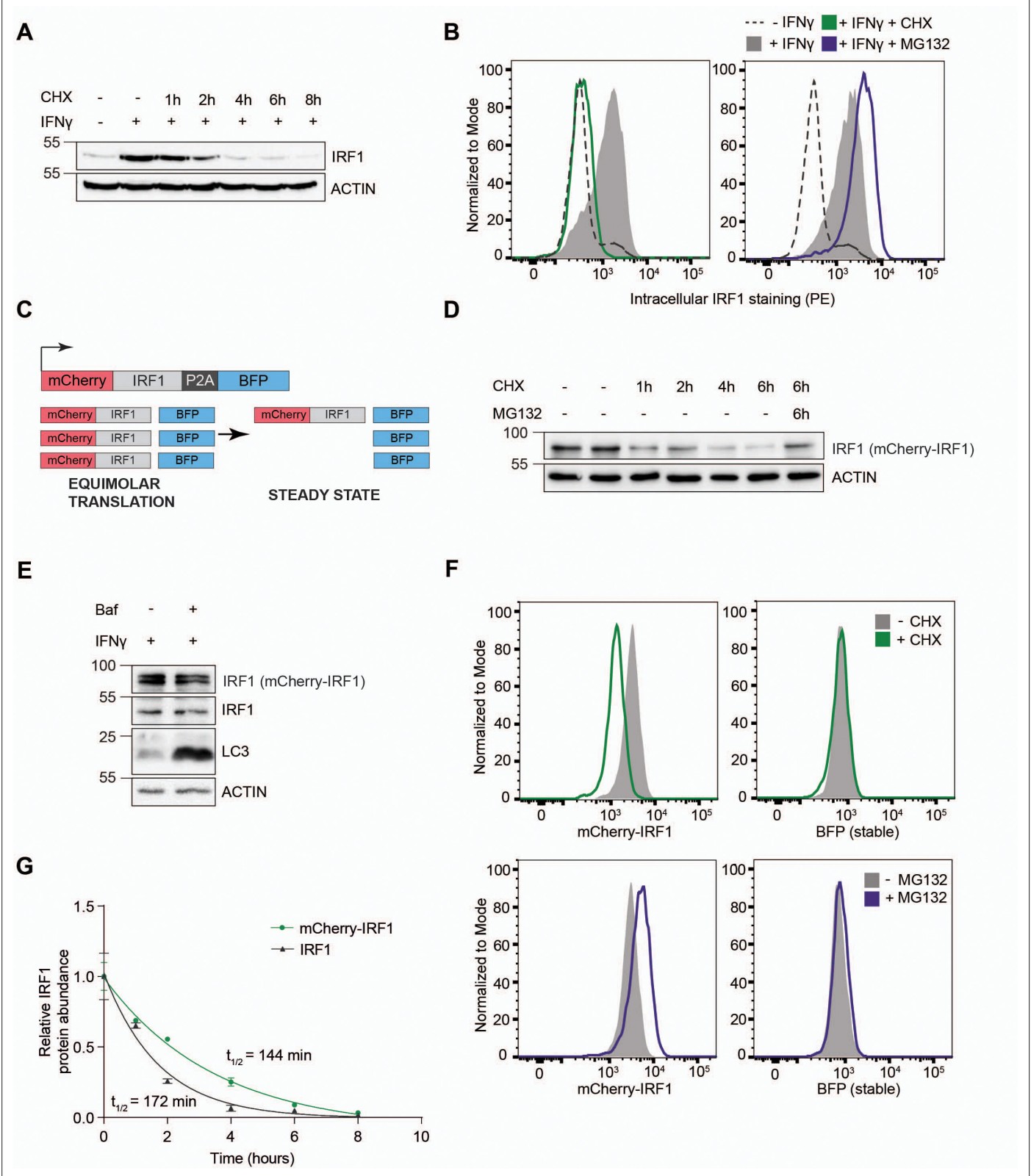

**Figure 1.** Establishment of an interferon regulatory factor 1 (IRF1) protein abundance reporter cell line for genetic screening. RKO cells were stimulated with IFNγ for 4 hr, and subsequently incubated with the translation inhibitor cycloheximide (CHX) in the absence or presence of proteasome inhibitor MG132, after which endogenous IRF1 levels were analyzed by (**A**) western blot or (**B**) intracellular staining and flow cytometry. (**C**) A dual-color reporter construct, from which unstable mCherry-IRF1 and stable BFP are synthesized in equimolar amounts, yet mCherry-IRF1 accumulates at low steady-state

*Figure 1 continued on next page*

*Figure 1 continued*

protein concentrations resulting from its proteasomal degradation. (**D**) RKO-mCherry-IRF1-P2A-BFP cells were incubated with CHX for the indicated times in the absence or presence of proteasome inhibitor MG132, after which mCherry-IRF1 levels were analyzed by western blot through detection of IRF1 protein. (**E**) RKO-mCherry-IRF1-P2A-BFP cells were stimulated with IFNγ for 4 hr, and subsequently incubated with lysosomal degradation inhibitor Bafilomycin A (Baf) for 4 hr, after which IRF1 levels were determined by western blot. (**F**) RKO-mCherry-IRF1-P2A-BFP cells were incubated with CHX or MG132 for 4 hr, after which mCherry-IRF and BFP protein levels were analyzed by flow cytometry. (**G**) RKO-mCherry-IRF1-P2A-BFP cells were incubated with CHX for the indicated times. Non-saturated western blot signals for mCherry-IRF1 and endogenous IRF1 protein were quantified, and normalized to ACTIN levels. Data represent means and s.d. (n=2 biological replicates). Half-life calculations were performed with GraphPad Prism (**v9**) by fitting the data to a one-phase decay nonlinear regression curve.

The online version of this article includes the following source data and figure supplement(s) for figure 1:

**Source data 1.** Western blots corresponding to *Figure 1A*.

**Source data 2.** Western blots corresponding to *Figure 1D*.

**Source data 3.** Western blots corresponding to *Figure 1E*.

**Figure supplement 1.** Establishment of an interferon regulatory factor 1 (IRF1) protein abundance reporter cell line for genetic screening.

**Figure supplement 1—source data 1.** Western blots corresponding to *Figure 1—figure supplement 1A*.

**Figure supplement 1—source data 2.** Western blots corresponding to *Figure 1—figure supplement 1C*.

with our findings by western blot readout, flow cytometry analysis also showed that both mCherry-IRF1 and mCherry-cMYC were degraded in the absence of de novo protein synthesis (*Figure 1F*, *Figure 1—figure supplement 1E*; top panels), and stabilized by proteasome inhibition (*Figure 1F*, *Figure 1—figure supplement 1E*; bottom panels), without affecting levels of the stable BFP internal control.

To determine IRF1 and cMYC half-lives, we quantified mCherry-IRF1 and IFNγ-induced endogenous IRF1 levels over time following inhibition of protein synthesis. Single exponential decay curve fitting indicated the half-life of endogenous IRF1 to be 72 min, whereas the half-life of mCherry-IRF1 was twice as long at 144 min (*Figure 1G*). We reasoned that the increased mCherry-IRF1 half-life would likely still allow identification of factors mediating degrading of endogenous IRF1, but could not rule out that some aspects of endogenous IRF1 degradation were differentially represented by mCherry-IRF1. Importantly, IFNγ treatment did not alter constitutively expressed mCherry-IRF1 levels (*Figure 1—figure supplement 1F*), indicating that its degradation machinery is likely already active under non-stimulated conditions. Similar measurements for mCherry-cMYC indicated a 24 min half-life for mCherry-cMYC and 19 min for endogenous cMYC in the same cells (*Figure 1—figure supplement 1C–D*), consistent with previously published work (*Welcker et al., 2004*; *Yada et al., 2004*). These combined data indicate that the N-terminal mCherry fusions of IRF1 and cMYC phenocopied the proteasomal degradation and instability of their endogenous counterparts.

Lastly, we assessed whether our selected monoclonal mCherry-IRF1 and mCherry-cMYC reporter cell lines had retained the ability to drive genome editing. To this end, cells were transduced with a lentiviral vector constitutively expressing an sgRNA targeting the cell-essential factor ribonucleotide reductase catalytic M1 (*RRM1*), while co-expressing iRFP. Subsequently, the ratio between these transduced cells and non-transduced cells in the same cell pool were measured over time by flow cytometry as a proxy for cell fitness.

As anticipated, the presence of Dox in the culture medium to induce Cas9 resulted in a near-complete loss of iRFP[+] sg*RRM1* cells from the cell pool in 6 days in both cell lines (*Figure 1—figure supplement 1G–H*), whereas parallel sg*AAVS1* control samples remained at a constant ratio. In contrast, in the absence of Dox, sg*RRM1* cells were not depleted over the 2-week course of the experiment. This indicated that the mCherry-IRF1 and mCherry-cMYC cell lines had minimal editing in the absence of Dox, yet efficient genome editing upon Dox-induced Cas9 expression.

Taken together, our data demonstrate that mCherry-IRF1 and mCherry-cMYC phenocopy the unstable nature of their endogenous counterparts and are likewise predominantly degraded by the proteasome. We concluded that the established reporter cell lines were suitable for subsequent genetic screening to identify IRF1 and cMYC stability regulators and that the established cell lines could be used for parallel screens to determine specificity of identified factors.

## Identification of the Cullin substrate receptor SPOP as a regulator of IRF1 abundance

The characterized mCherry-IRF1 and mCherry-cMYC reporter cell lines were subsequently used for genome-wide genetic screening for regulators of their protein abundance. To this end, a genome-wide lentiviral sgRNA library was integrated in the cell lines at a low multiplicity of infection (MOI) to ensure the targeting of one gene per cell (*Figure 2A*, *Figure 2—figure supplement 2A*), followed by selection for G418 resistance to enrich for sgRNA-expressing cells (*Figure 2—figure supplement 2B–C*). At 3 and 6 days after Cas9 induction with Dox, the top 1–3% of cells with the highest and lowest mCherry levels were collected by flow cytometry (*Figure 2A*, *Figure 2—figure supplement 3*, *Figure 2—figure supplement 4*), to enrich for cells with decreased or increased levels of the unstable mCherry-IRF1 or mCherry-cMYC. In parallel, the top 1–3% of cells with the highest and lowest levels of the stable BFP control were collected to identify non-specific effects (*Figure 2A*, *Figure 2—figure supplement 3*, *Figure 2—figure supplement 4*).

The two harvesting time points were previously empirically determined to be optimal for capturing essential genes (day 3), and genes that encode proteins whose levels require longer times to deplete (day 6) (*Figure 1—figure supplement 1G–H*; *de Almeida et al., 2021*). Integrated sgRNA coding sequences from the selected cell pools were quantified by next-generation sequencing (NGS) and compared to relative frequencies determined in unsorted cells representing the initial library (*Figure 2—figure supplement 5A*). The robustness and uniformity of the screens in both cell lines is underpinned by the fact that all sgRNA coding sequences in unsorted samples were covered by 100–1000 reads (*Figure 2—figure supplement 5B–C*).

As anticipated, the mCherry-cMYC screen strongly identified its well-known degrader FBXW7 in mCherry-cMYC$^{high}$ cells (*Figure 2B*, *Figure 2—figure supplement 1A–D*). Moreover, it identified other UPS components reported to mediate cMYC degradation (*Figure 2—figure supplement 1A–D*), such as proteasome components (PSMC5, PSMD4, PSMD11), the proteasome regulator AKIRIN2 (*de Almeida et al., 2021*), and the UPS-related ATPase VCP/p97 (*Heidelberger et al., 2018*). These factors were not enriched in the BFP$^{high}$ stable control cells (*Figure 2—figure supplement 1C–D*), attesting to the specificity and identification power of the experimental system.

The mCherry-IRF1 screen identified 27 high confidence factors which specifically affected mCherry-IRF1 abundance, but neither mCherry-cMYC nor the stable BFP controls (*Figure 2C–E*). The only candidate strongly enriched at both 3 and 6 days after Cas9 induction was the Cullin E3 ligase receptor speckle type BTB/POZ protein (SPOP), which did not score in the mCherry-cMYC screen (*Figure 2B–E*, *Figure 2—figure supplement 1E–F*). In addition, seven nuclear transport-related factors (NUP35, NUP85, NUP93, NUP155, NUP160, TMEM48, NUTF2) were identified (*Figure 2E*, *Figure 2—figure supplement 1F*), suggesting that nuclear translocation of IRF1 or one of its degraders may be important for its degradation. Consistent with this notion, SPOP has been previously reported to localize to the nucleus and degrade nuclear substrates (*Marzahn et al., 2016*; *Usher et al., 2021*). Moreover, a chromatin reader component of the NuA4 histone acetyltransferase complex (YEATS4) and two factors with limited functional annotation (GPATCH1, C14ORF102) were identified (*Figure 2E*).

Previously described regulators of IRF1 proteasomal turnover such as FBXW7, MDM2, or STUB1/CHIP (*Garvin et al., 2019*; *Landré et al., 2013*; *Narayan et al., 2011b*) did not score as significant IRF1 regulators in our screen, indicating that they are unlikely to play a significant role in IRF1 degradation in this cellular context, although we cannot rule out that unknown technical aspects in the screen prevented their identification. Importantly, FBXW7 was identified as the strongest factor mediating cMYC degradation (*Figure 2B*, *Figure 2—figure supplement 1A–D*), underpinning the functionality and specificity of the screening approach. In summary, our genetic screen identified the E3 ligase substrate receptor SPOP as a specific candidate to regulate IRF1 protein abundance.

## SPOP regulates IRF1 protein levels

To determine whether our genetic screen results were predictive, individual genes identified in cell pools with increased mCherry-cMYC or mCherry-IRF1 levels were targeted. Ablation of top-scoring putative regulators of cMYC (*FBXW7*, *UBE3C*, *VCP*) substantially increased mCherry-cMYC levels (*Figure 3—figure supplement 1A–C*), whereas targeting SPOP specifically increased the abundance of mCherry-IRF1 (*Figure 3A*).

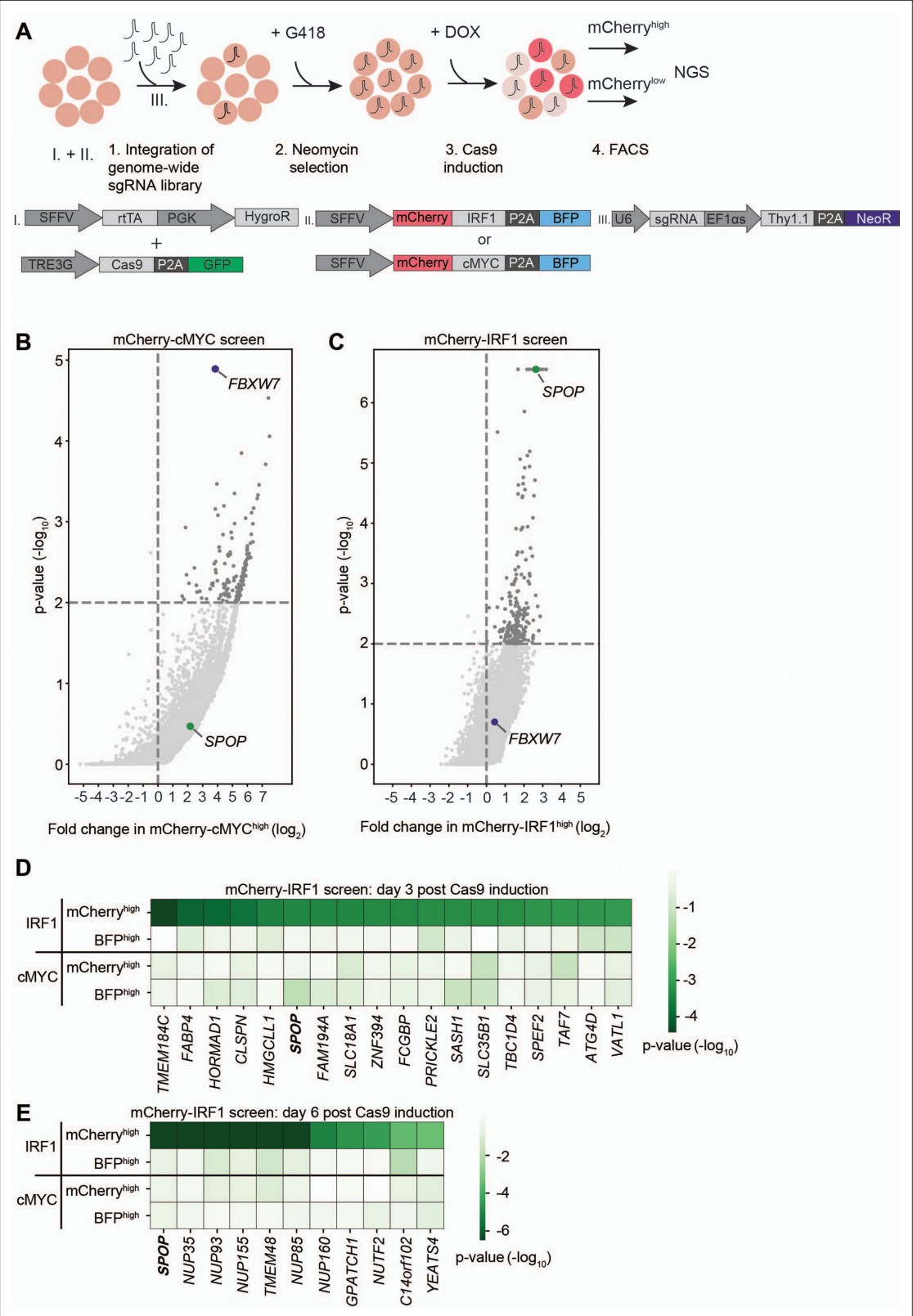

**Figure 2.** Identification of SPOP as a regulator of interferon regulatory factor 1 (IRF1) abundance. (**A**) Overview of fluorescence-activated cell sorting (FACS)-based CRISPR-Cas9 knockout screening procedure using the RKO-mCherry-IRF1-P2A-BFP or RKO-mCherry-cMYC-P2A-BFP cell lines. Cells expressing high and low levels of mCherry-cMYC or mCherry-IRF1 protein were sorted and their integrated sgRNA coding sequences determined by next-generation sequencing. (**B–C**) Read counts per million in the mCherry-cMYC[high] and mCherry-IRF1[high] cells at 6 days after Cas9 induction were

*Figure 2 continued on next page*

*Figure 2 continued*

compared to those in unsorted cells from the same day, sgRNA enrichment calculated by MAGeCK analysis, and log$_2$-fold change and adjusted p-value plotted. Specific top hits SPOP and FBXW7 are shown for the (**B**) mCherry-cMYC or (**C**) mCherry-IRF1 screen. (**D–E**) 20 genes enriched in the mCherry-IRF1$^{high}$ sorted populations with lowest adjusted p-values were selected and any genes enriched in BFP$^{high}$ or mCherry$^{low}$ populations were excluded. Screen-specific adj. p-value heatmaps were generated for mCherry-IRF1$^{high}$ cells at (**D**) 3 days or (**E**) 6 days post-Cas9 induction.

The online version of this article includes the following figure supplement(s) for figure 2:

**Figure supplement 1.** Identification of SPOP as a regulator of interferon regulatory factor 1 (IRF1) abundance.

**Figure supplement 2.** Identification of SPOP as a regulator of interferon regulatory factor 1 (IRF1) abundance.

**Figure supplement 3.** Identification of SPOP as a regulator of interferon regulatory factor 1 (IRF1) abundance.

**Figure supplement 4.** Identification of SPOP as a regulator of interferon regulatory factor 1 (IRF1) abundance.

**Figure supplement 5.** Identification of SPOP as a regulator of interferon regulatory factor 1 (IRF1) abundance.

Next, we set out to assess whether SPOP regulates endogenous IRF1 protein levels. To this end, cells were transduced with sg*SPOP* expression vectors, Cas9 was induced with Dox, and their endogenous intracellular IRF1 and cMYC protein levels determined by flow cytometry (*Figure 3B–D*). Consistent with our screen data (*Figure 2*, *Figure 2—figure supplement 1*), *SPOP* ablation increased endogenous IRF1 levels by 40%, whereas endogenous cMYC protein levels were not affected (*Figure 3C*).

The sgRNA library design with coverage of each gene by six sgRNAs made it unlikely that off-target effects of the *SPOP*-targeting sgRNAs were responsible for the selected phenotype of increased mCherry-IRF1. However, to measure whether individual sgRNAs would result in the same IRF1-stabilizing phenotype, we transduced cells individually with seven different, independent, sg*SPOP* sequences, and analyzed intracellular IRF1 levels by flow cytometry (*Figure 3D*). All seven different sgRNAs significantly increased IRF1 levels compared to the two independent controls, indicating that specific *SPOP* targeting underlies this phenotype, and not off-target effects (*Figure 3D*). Consistent with this notion, western blot analysis of three independent monoclonal *SPOP* knockout cell lines showed that IRF1 levels were increased by 50–70% (*Figure 3E–F*). This effect of *SPOP* ablation specifically affected IRF1 protein levels, as *IRF1* mRNA concentrations were not significantly changed in *SPOP* knockout cells (*Figure 3—figure supplement 1D–E*).

Lastly, we assessed the importance of SPOP for IRF1 degradation in non-cancer-derived cells, and across species. In line with our results from human cell lines, *SPOP* ablation in mouse embryonic fibroblasts (MEFs) likewise increased endogenous IRF1 protein levels (*Figure 3G–H*), indicating a conserved mechanism across species and non-transformed cell types. In summary, these findings demonstrate that SPOP is important for regulating IRF1 protein levels, without affecting its mRNA concentrations.

## SPOP targets S/T-rich degrons in IRF1

Having identified SPOP in a phenotypic genetic screen, it was possible that the effects of *SPOP* ablation on IRF1 were direct, or indirect through unknown other mediators. As a first step to investigate a possible direct targeting of IRF1 by SPOP, we first analyzed their subcellular distribution to determine whether they reside in the same compartment.

To this end, HeLa and RKO cells expressing exogenous IRF1 and SPOP were analyzed by confocal immunofluorescence microscopy. Consistent with previous reports, both IRF1 and SPOP localized in the nucleoplasm (*Figure 4A*), indicating that they are present in the same subcellular compartment, and thus could interact. While both proteins localized dispersedly in the nucleoplasm, in both tested cell types SPOP was also present at nuclear speckles, as previously described (*Marzahn et al., 2016*).

We reasoned that the steady-state levels of the SPOP-targeted IRF1 pool could be low under these conditions of ongoing IRF1 degradation. To stabilize the actively targeted IRF1 protein pool, cells were treated with proteasome inhibitor. Under these conditions, the diffuse nucleoplasmic IRF1 signal increased (*Figure 4—figure supplement 1A*, *Figure 4—figure supplement 2*), suggesting that this represents the unstable cellular IRF1 pool. Moreover, in the presence of proteasome inhibitor, the SPOP puncta increased in size and signal intensity, consistent with previous reports that these are sites of SPOP-dependent degradation of a subset of its targets (*Marzahn et al., 2016*; *Usher et al., 2021*). However, none of the IRF1 signal was detected in the SPOP puncta (*Figure 4—figure supplement*

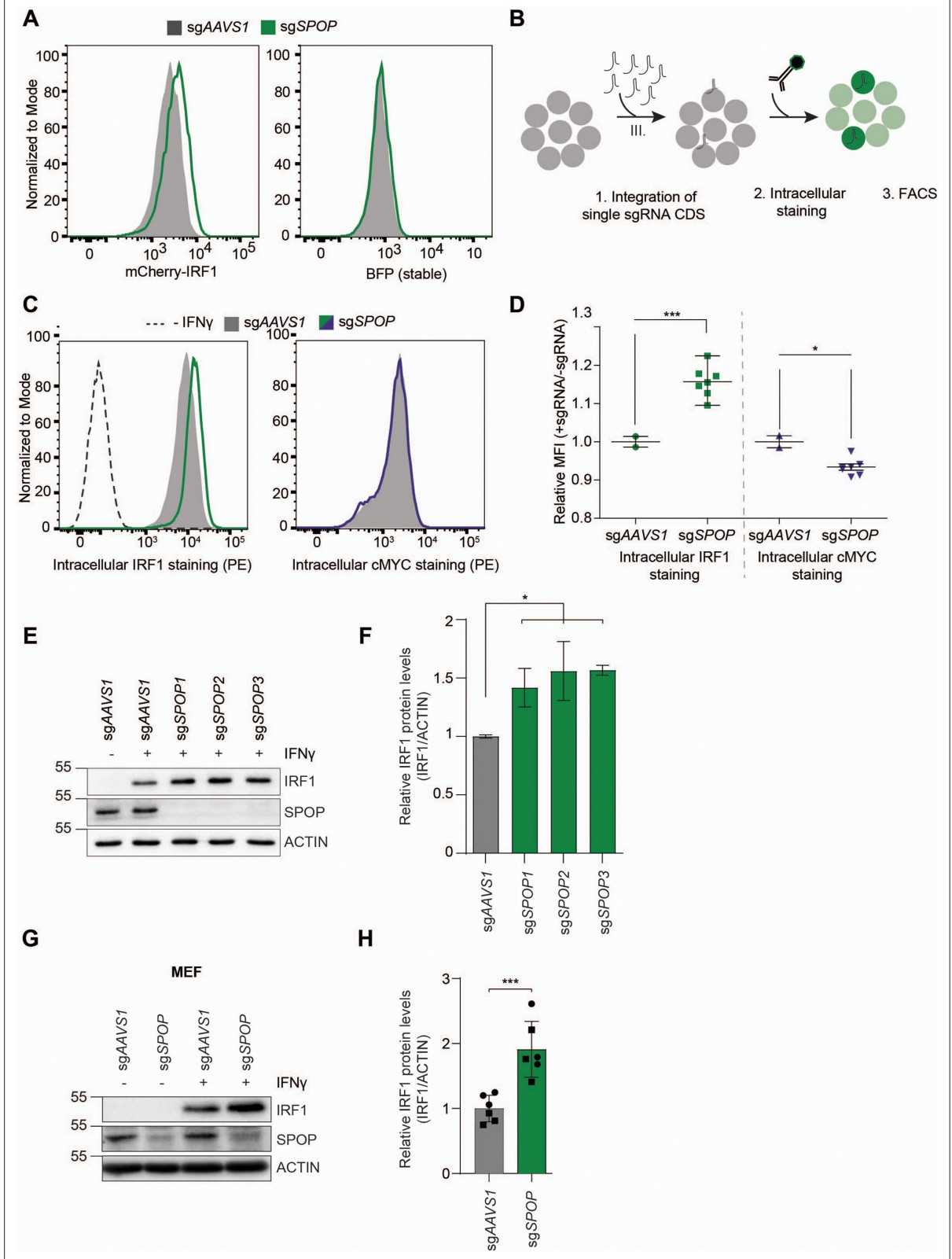

**Figure 3.** SPOP regulates interferon regulatory factor 1 (IRF1) protein levels. (**A**) RKO-mCherry-IRF1-P2A-BFP cells expressing sg*AAVS1* or sg*SPOP* were treated with Dox to induce Cas9. Subsequently, mCherry-IRF1 and BFP protein levels were analyzed by flow cytometry. (**B**) RKO-Cas9-P2A-GFP cells expressing sg*AAVS1* or sg*SPOP* were treated with Dox to induce Cas9, treated with IFNγ for 4 hr, after which endogenous IRF1 or cMYC levels were determined by (**C–D**) intracellular staining and flow cytometry. In (**D**), each data point represents targeting with a different, independent sg*SPOP*

*Figure 3 continued on next page*

*Figure 3 continued*

sequence. Data were analyzed by unpaired two-tailed t-test; n=2 biological replicates. *p<0.05, ***p<0.001. (**E**) Independent monoclonal RKO-Cas9-GFP cell lines with wt *IRF1* alleles derived from sg*AAVS1* expressing cells, or homozygous *SPOP* disrupted alleles were stimulated with IFNγ for 4 hr, after which their IRF1 and SPOP protein levels were determined by western blot (WB), and (**F**) quantified (unpaired t-test, n=2). (**G**) Mouse embryonic fibroblast (MEF) cells were treated with IFNγ for 4 hr, after which endogenous IRF1 levels were analyzed by WB and (**H**) quantified (unpaired t-test, n=6 biological replicates from two independently established MEF cell lines indicated by circle and square symbols), ***p<0.001.

The online version of this article includes the following source data and figure supplement(s) for figure 3:

**Source data 1.** Western blots corresponding to *Figure 3E*.

**Source data 2.** Western blots corresponding to *Figure 3G*.

**Figure supplement 1.** SPOP regulates interferon regulatory factor 1 (IRF1) protein levels.

*1A*). Together, these data suggest that IRF1 is likely targeted for degradation in the nucleoplasm, and that this could potentially be mediated by the diffuse nucleoplasmic SPOP pool.

To further extend these analyses, subcellular fractionation of *AAVS1-* and *SPOP*-targeted cells was performed. Consistent with our microscopy results (*Figure 4A*, *Figure 4—figure supplement 1A*), IFNγ treatment rapidly induced IRF1 in the nuclear fraction, which even further increased upon *SPOP* depletion (*Figure 4B*, *Figure 4—figure supplement 1B–C*). These data reinforce the conclusion that SPOP targets IRF1 in the nucleus, where it is subsequently degraded.

Previous studies had identified an S/T-rich SPOP binding consensus of $[\Phi]$-$[\Pi]$-$[S]$-$[S/T]$-$[S/T]$, in which $\Phi$ and $\Pi$ are non-polar and polar residues, respectively (*Zhuang et al., 2009*). We reasoned that the presence of any sequences matching this SPOP degron in IRF1 could mediate SPOP binding to IRF1, thereby targeting it for ubiquitination and degradation.

A multiple sequence alignment of IRF1 across species (*Figure 4—figure supplement 3*) identified four sequences in IRF1 matching the SPOP degron consensus (*Figure 4C*), consistent with predictions (ScanProsite – Expasy, and *Hou et al., 2022*). One of the putative SPOP degrons was identified in the region between the DNA-binding domain and the IAD2 domain, whereas the remaining three were predicted to be inside the IAD2 domain, which is important for transactivation and dimerization (*Figure 4C*, *Figure 4—figure supplement 3*).

To test whether these predicted degron sites in IRF1 are important for SPOP-dependent degradation, an IRF1 mutant was generated in which the core S/T residues in each putative SPOP binding domain (SBD) were mutated to A residues (IRF1-SBD$_{mut}$). Subsequently, wild-type IRF1 or IRF1-SBD$_{mut}$ were expressed in the presence of increasing amounts of a wild-type SPOP expression plasmid (*Figure 4D*). SPOP expression decreased wtIRF1 protein levels in a dose-dependent manner (*Figure 4D*), without affecting the stable EGFP internal control. In contrast, protein levels of the IRF1 degron mutant remained unaffected by exogenous SPOP expression, suggesting that one or more of the four predicted degrons are required for SPOP-dependent IRF1 degradation.

Next, we set out to investigate whether IRF1 and SPOP interact in cells and whether this interaction is mediated through the predicted IRF1 S/T degrons. To directly assess complex formation between SPOP and IRF1, SPOP was immunoprecipitated from cells expressing wtIRF1 or IRF1-SBD$_{mut}$. Subsequent western blot analysis for IRF1 identified wtIRF1, but not IRF1-SBD$_{mut}$ (*Figure 4E*). The reciprocal approach in which wtIRF1 or IRF1-SBD$_{mut}$ were immunoprecipitated, exclusively identified SPOP upon purification of wtIRF1, but not its degron mutant (*Figure 4F*).

SPOP interaction with S/T degrons has been previously mapped to its MATH domain (*Zhuang et al., 2009*). In line with this observation, wtIRF1 interacted with wtSPOP, yet failed to co-immunoprecipitate MATH-domain mutated SPOP (SPOP-MATH$_{mut}$) (*Figure 4G*). Consistently, the reciprocal pull-down of wtSPOP identified wtIRF1, whereas the SPOP mutant showed strongly reduced interaction with IRF1 (*Figure 4H*).

In line with one or more of the four identified degrons being important for SPOP-dependent IRF1 turnover, IRF1 ubiquitination was significantly increased upon co-expression of wtSPOP, but not with MATH-domain mutated SPOP (SPOP-MATH$_{mut}$). Upon mutation of the SPOP binding sequences, IRF1 ubiquitination was significantly decreased (*Figure 4I–J*). From this, we concluded that some or all of the SPOP degrons are required for IRF1 poly-ubiquitination, although the remaining ubiquitination in the IRF1 mutant indicates that additional E3 ligases may ubiquitinate IRF1 as well. It remains to be

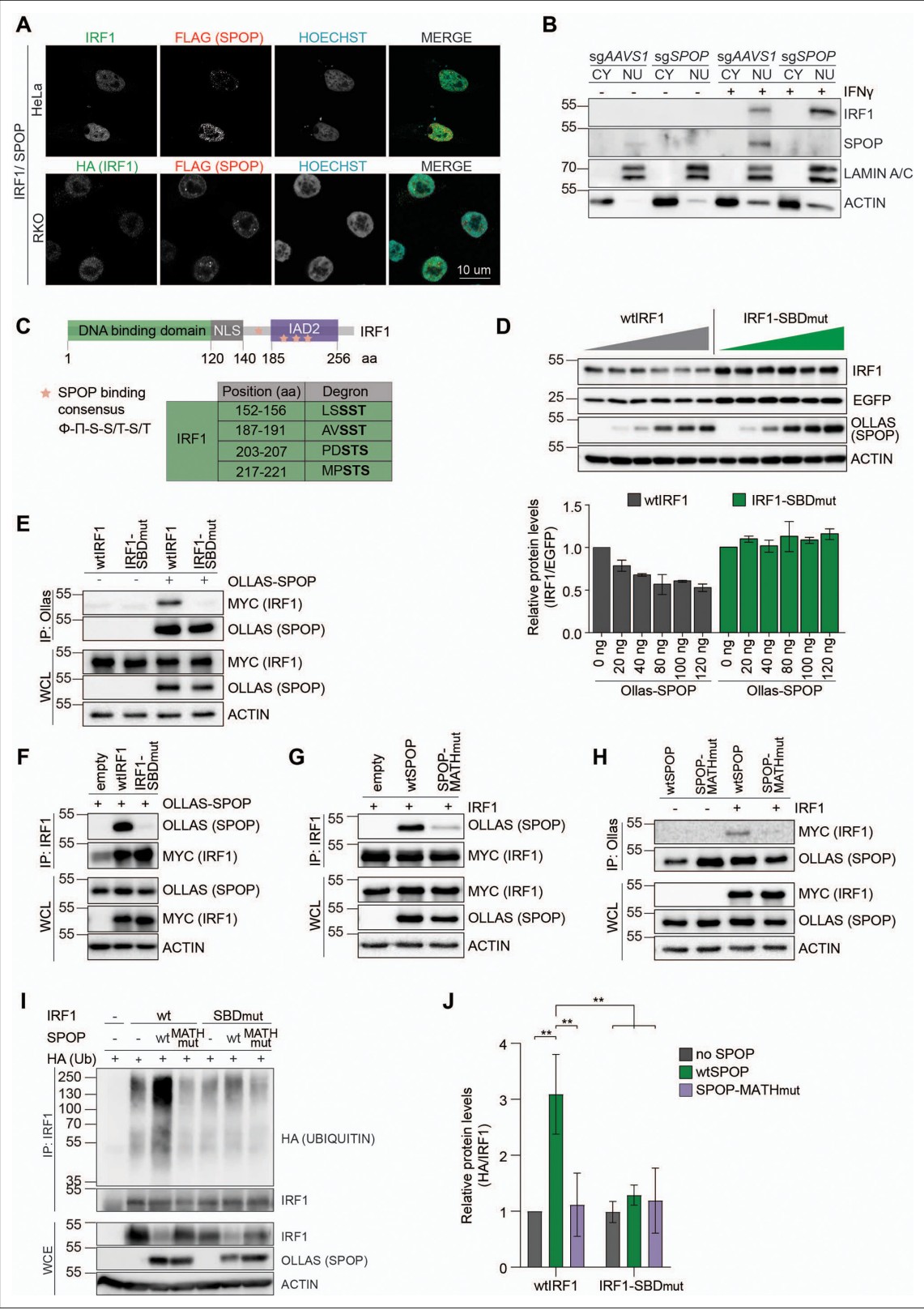

**Figure 4.** SPOP targets S/T-rich degrons in interferon regulatory factor 1 (IRF1). (**A**) Representative images of HeLa cells, transfected with expression plasmids for MYC-tagged IRF1 and FLAG-tagged SPOP, and RKO cells, stably expressing HA-tagged IRF1 and FLAG-tagged SPOP. At 48 hr post-transfection (HeLa) or 48 hr after seeding (RKO), cells were fixed, stained for the indicated factors, and their localization analyzed by confocal immunofluorescence microscopy. (**B**) Independent monoclonal RKO-Cas9-GFP cells lines with wt *IRF1* alleles derived from sg*AAVS1* expressing cells,

*Figure 4 continued on next page*

*Figure 4 continued*

or homozygous *SPOP* disrupted alleles were stimulated with IFNγ for 4 hr, after which whole cell (WCL), cytosolic (CY), and nuclear (NU) fractions were collected. IRF1 and SPOP protein levels were determined by western blot. (**C**) Schematic representation of human IRF1 and its conserved domains. The SPOP consensus binding sequence is displayed and matching degron sequences in human IRF1 indicated. (**D**) HEK-293T cells were co-transfected with expression plasmids for EGFP, Ollas-tagged wtSPOP, and either wtIRF1 or an IRF1 mutant in which the S/T residues in the four indicated SPOP binding domains (SBD) were mutated to A (IRF1-SBD$_{mut}$). At 36 hr post-transfection, cell lysates were analyzed by western blot and quantified. Data represent means and s.d. (n=3). (**E–H**) HEK-293T cells were transfected with the indicated expression plasmids for wtIRF1, IRF1-SBD$_{mut}$, Ollas-tagged wtSPOP, or an SPOP mutant in which its substrate interaction site in its MATH domain was mutated (SPOP-MATH$_{mut}$). At 36 hr post-transfection, cells were stimulated with epoxomicin (EPOX) for 4 hr, and subsequently interaction was analyzed by co-IP. (**I–J**) HEK-293T cells were transfected with the indicated plasmids encoding MYC-tagged wtIRF1 or IRF1-SBD$_{mut}$, Ollas-tagged wtSPOP and SPOP-MATH$_{mut}$, as well as HA-tagged ubiquitin. 36 hr post-transfection, cells were treated with EPOX for 4 hr, and subsequently MYC-tagged IRF1 was immunoprecipitated and (**I**) ubiquitination analyzed by western blot, and (**J**) quantified by densitometry. Data were analyzed by two-way ANOVA; n=3 biological replicates, data represent means and s.d., **p<0.01.

The online version of this article includes the following source data and figure supplement(s) for figure 4:

**Source data 1.** Western blots corresponding to *Figure 4A*.

**Source data 2.** Western blots corresponding to *Figure 4B*.

**Source data 3.** Western blots corresponding to *Figure 4D*.

**Source data 4.** Western blots corresponding to *Figure 4E*.

**Source data 5.** Western blots corresponding to *Figure 4F*.

**Source data 6.** Western blots corresponding to *Figure 4G*.

**Source data 7.** Western blots corresponding to *Figure 4H*.

**Source data 8.** Western blots corresponding to *Figure 4I*.

**Figure supplement 1.** SPOP targets S/T-rich degrons in interferon regulatory factor 1 (IRF1).

**Figure supplement 1—source data 1.** Western blots corresponding to *Figure 4—figure supplement 1A*.

**Figure supplement 1—source data 2.** Western blots corresponding to *Figure 4—figure supplement 1B*.

**Figure supplement 1—source data 3.** Western blots corresponding to *Figure 4—figure supplement 1I*.

**Figure supplement 2.** Identification of SPOP as a regulator of interferon regulatory factor 1 (IRF1) abundance.

**Figure supplement 2—source data 1.** Western blots corresponding to *Figure 4—figure supplement 2*.

**Figure supplement 3.** Identification of SPOP as a regulator of interferon regulatory factor 1 (IRF1) abundance.

determined whether these SPOP-independent poly-ubiquitin chains have a topology associated with proteasomal degradation.

Subsequently, we set out to identify the lysine residues required for IRF1 turnover. To this end we expressed SPOP together with full-length IRF1, in which all lysines in only the indicated protein domains were mutated to arginines to prevent ubiquitination (*Figure 4—figure supplement 1D–E*). Mutation of all lysines in the N-terminal half of the DNA-binding domain (DBD1), its C-terminal half (DBD2), the NLS region, or its C-terminal domain (C-term.) did not alter its sensitivity to treatment with proteasome inhibitor (*Figure 4—figure supplement 1D–E*). In contrast, an IRF1 mutant in which all lysines were mutated in the entire IRF1 protein (allKtoR) was no longer stabilized by proteasome inhibition, suggesting that SPOP targets lysines in multiple IRF1 domains.

Lastly, we tested whether wtSPOP or SPOP-MATH$_{mut}$ could rescue IRF1 degradation in *SPOP* KO cells. To this end, expression plasmids for wtSPOP or its MATH domain mutant counterpart were delivered to *SPOP* KO cells, and endogenous IRF1 levels measured by flow cytometry (*Figure 4—figure supplement 1F*). In line with previous observations, knockout of *SPOP* increased IRF1 protein levels (*Figure 4—figure supplement 1G*). Expression of wtSPOP in *AAVS1*-targeted RKO cells did not alter endogenous IRF1 protein concentrations compared to an empty vector control (*Figure 4—figure supplement 1H*, left panel), indicating that SPOP levels in these cells were likely not limiting for IRF1 turnover. In contrast, the elevated IRF1 protein levels in *SPOP* KO cells were decreased by exogenous wtSPOP expression to levels measured in wild-type cells, whereas they were not affected by the empty vector control (*Figure 4—figure supplement 1H*; middle panel and *Figure 4—figure supplement 1I*). Consistent with our interaction data and requirement of the SPOP MATH domain for IRF1 targeting (*Figure 4G–H*), IRF1 levels were only rescued by wtSPOP expression, but not by its MATH domain mutant counterpart (*Figure 4—figure supplement 1H*, right panel).

Taken together, these data indicate that IRF1 is targeted by SPOP for ubiquitination through interaction of the SPOP MATH domain with S/T-rich degrons in IRF1, resulting in subsequent IRF1 degradation.

## SPOP controls IRF1-dependent cellular output

IRF1 is required for efficient transcription of interferon-induced genes (ISGs). IRF1 is in particular important for transcriptional induction of a subset of ISGs, such as members of the guanylate binding protein (GBP) family in response to IFNs (*Ramsauer et al., 2007*). By querying the harmonizome project for IRF1 transcriptional targets (*Rouillard et al., 2016*), we identified the ISGs *GBP2*, *GBP3*, and *IFIT3* as putatively IRF1-dependent immune effector ISGs, and *PARP14* and *APOL2* as IRF1-dependent non-immune effector genes in RKO cells. *NLRC5* was included as an IRF1-independent control gene. We asked whether SPOP-dependent control of IRF1 protein abundance influences IRF1-dependent transcriptional output. To this end, mRNA levels of IRF1 target genes were measured by RT-qPCR in either control cells, *SPOP*-deficient, *IRF1*-deficient, or *IRF1/SPOP* double knockout cells (*Figure 5A–B*, *Figure 5—figure supplement 1A*).

As expected, *SPOP* ablation did not significantly change the mRNA concentration of IRF1 independently transcribed *NLRC5* (*Figure 5A*). In contrast, *SPOP* ablation significantly increased the mRNA levels of IRF1-dependent immune-related targets *GBP2*, *GBP3*, and *IFIT3* (*Figure 5A*) in the presence of IRF1. Expression was reduced to equal levels in both *IRF1* and *IRF1/SPOP* knockout cells, indicating that differences in ISG expression in *SPOP* KO cells are dependent on IRF1-driven transcription. Moreover, non-immune transcripts *APOL2* and *PARP14* (*Figure 5B*) were likewise increased in the absence of SPOP in an IRF1-dependent manner.

To further investigate the effect of SPOP on IRF1-dependent transcriptional output, the effect of exogenous SPOP expression in a reporter assay was assessed. To this end, wtIRF1 or IRF1-SBD$_{mut}$ were co-expressed with an IFN-stimulated response element (ISRE) luciferase reporter (*Figure 5C*). wtIRF1 expression increased reporter activation by 408-fold (*Figure 5C*). Consistent with our findings that the IRF1-SBD$_{mut}$ is expressed at increased steady-state levels (*Figure 4D*), expression of the same amount of plasmid encoding this mutant resulted in an 85% increase in ISRE-reporter activity (*Figure 5C*). In line with SPOP degron-dependent degradation, wtIRF1-driven ISRE activation was reduced 11-fold by SPOP co-expression, whereas its degron-mutated IRF1-SBD$_{mut}$ counterpart was unaffected (*Figure 5C*).

To exclude any confounding effects, we reduced the amount of IRF1-SBD$_{mut}$ expression construct by half to match the steady-state protein levels of wtIRF1. Likewise, under these conditions wtSPOP expression specifically reduced wtIRF1-driven reporter expression, but not its IRF1-SBD$_{mut}$ counterpart (*Figure 5D*). Moreover, IRF1-driven reporter activation was only reduced by wtSPOP, but not SPOP-MATH$_{mut}$, which cannot interact with IRF1 (*Figure 5D*). From these data, we concluded that one or more degrons in the IRF1 substrate as well as the substrate binding domain in SPOP are required for reducing IRF1 levels, thereby controlling its transcriptional output.

Unlike other IRFs, IRF1 also drives the expression of various cell cycle inhibiting factors (e.g. p21) (*Armstrong et al., 2012*; *Tanaka et al., 1996*), making it an important tumor suppressor (*Nozawa et al., 1999*; *Xie et al., 2003*). Therefore, we next determined whether *SPOP* loss would affect cell fitness in a competition assay. To avoid any confounding and desensitization effects of continuous IFNγ treatment to upregulate IRF1, we targeted SPOP in our mCherry-IRF1 screening cell line, and analyzed their relative cell fitness in a mixed culture with untransduced cells (*Figure 5E*). Over the course of 20 days, targeting *SPOP* in cells not exogenously expressing mCherry-IRF1 (wt) reduced fitness by 15%, whereas this was further reduced by nearly twofold in mCherry-IRF1-expressing cells, relative to their respective sg*AAVS1* controls (*Figure 5E*). From these data we concluded that *SPOP* loss disproportionally reduces cell fitness in the presence of IRF1, consistent with increased IRF1 protein concentrations in the absence of SPOP.

Lastly, we addressed the biological relevance of *SPOP* loss on the cellular antiviral state. Pre-treatment of both sg*AAVS1* and sg*SPOP* cells with IFNγ reduced vesicular stomatitis virus (VSV) progeny virus production by approximately fivefold in a multi-cycle infection experiment (*Figure 5—figure supplement 1B*). However, progeny virus titers were not significantly different from sg*AAVS1* control cells (*Figure 5—figure supplement 1B*).

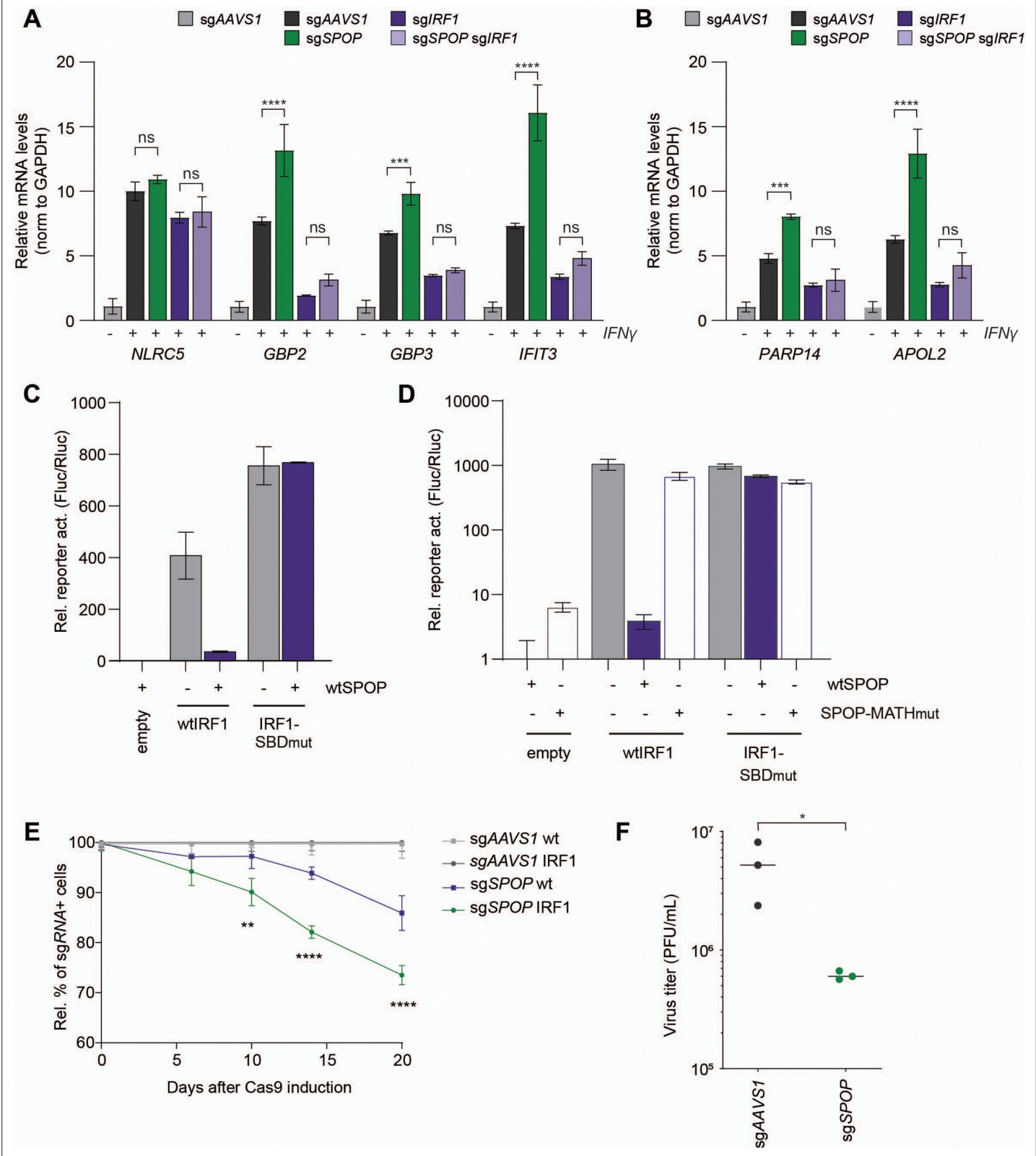

**Figure 5.** SPOP controls interferon regulatory factor 1 (IRF1)-dependent cellular output. RKO cells expressing either sg*AAVS1*, sg*SPOP*, sg*IRF1*, or sg*SPOP* and sg*IRF1* together were stimulated with IFNγ for 4 hr and concentrations of the indicated (**A**) immune-related or (**B**) non-immune-related mRNAs analyzed by RT-qPCR (n=2 biological replicates). Data represent means and s.d. Data were analyzed by unpaired two-tailed t-test. *p<0.05, ***p<0.001. (**C–D**) HEK-293T cells were transfected with expression plasmids for an ISRE-Fluc reporter, and (**C**) equal amounts of the indicated IRF1

*Figure 5 continued on next page*

*Figure 5 continued*

and SPOP expression constructs, or (**D**) half of the amount of IRF1-SBD$_{mut}$ plasmid to match IRF1 protein levels. After 24 hr cell lysates were analyzed by dual luciferase assay. n=3 biological replicates. Data represent means and s.d. (**E**) RKO-mCherry-IRF1-P2A-BFP cells or the parental cell line with only dox-inducible Cas9-GFP (wt) were transduced with iRFP vectors expressing sg*AAVS1* or sg*SPOP* in 30–60% of all cells. Gene editing was induced with Dox and the fraction of transduced cells monitored for 20 days. At the indicated times, the fraction of sgRNA vector-positive cells was determined by flow cytometry. Graphs show relative abundance of sg*SPOP*-positive cells normalized to sg*AAVS1* of the same cell line on day 0. Data were analyzed by two-way ANOVA; n=3 biological replicates; **p<0.01, ****p<0.0001. (**F**) RKO-mCherry-IRF1-P2A-BFP cells in which *IRF3* and *STAT1* were targeted by sgRNAs, were transduced with expression vectors for either sg*AAVS1* or sg*SPOP*. sgRNA-positive cells were sorted by fluorescence-activated cell sorting (FACS), and subsequently infected with vesicular stomatitis virus (VSV) at a multiplicity of infection (MOI) of 0.0005. After 26 hr, supernatants were collected, and progeny virus titers determined by plaque assays.

The online version of this article includes the following source data and figure supplement(s) for figure 5:

**Figure supplement 1.** SPOP controls interferon regulatory factor 1 (IRF1)-dependent cellular output.

**Figure supplement 1—source data 1.** Western blots corresponding to *Figure 5—figure supplement 1A*.

**Figure supplement 1—source data 2.** Western blots corresponding to *Figure 5—figure supplement 1C*.

Since IRF1 drives only the transcription of a subset of ISGs in IFNγ-stimulated cells, we reasoned that most of the antiviral state in cells could be conferred by the STAT1 homo-dimeric transcription factor complex, in an IRF1-independent manner. Therefore, we next performed VSV infections in cells exogenously expressing mCherry-IRF1, in which most of the antiviral state is IRF1-dependent (*Schoggins et al., 2011*). To further limit IRF1-independent ISG induction during infection, sgRNAs targeting *IRF3* and *STAT1* were expressed (*Figure 5—figure supplement 1C*; *Ramsauer et al., 2007*; *Schoggins et al., 2011*), in combination with either sg*AAVS1* or sg*SPOP*. sg*AAVS1*-expressing mCherry-IRF1 cells produced approximately 10-fold less progeny virus (*Figure 5F*), compared to its counterparts not expressing IRF1 (*Figure 5—figure supplement 1B*), indicating that mCherry-IRF1 expression conferred an antiviral effect. Consistent with increased IRF1 protein levels, loss of *SPOP* further reduced VSV progeny titers by approximately 10-fold (*Figure 5F*). These results indicate that SPOP plays an important role in limiting intracellular IRF1 protein concentrations, and that its loss increases IRF1-dependent cell fitness (*Figure 5E*) and antiviral immunity (*Figure 5F*).

## Discussion

The importance of IRF1 in myeloid cells for driving CXCL9 and CXCL10 expression and thereby being broadly critical for their innate immune response has been described (*Forero et al., 2019*). Moreover, IRF1 is a critical tumor suppressor, which is underpinned by the observation that IRF1 is mutated or lost in many cancers (*Nozawa et al., 1999*; *Xie et al., 2003*). Some ubiquitin E3 ligases controlling IRF1 protein turnover have been described, such as FBXW7, MDM2, and STUB1 (*Garvin et al., 2019*; *Landré et al., 2013*; *Narayan et al., 2011b*). Their differential importance for regulating IRF1 turnover in different cell types and tissues draws an interesting functional parallel with cMYC, which is also targeted for degradation by cell-type-specific E3 ligases (*Sun et al., 2021*).

Here, we show that the Cullin E3 ligase receptor SPOP is important for IRF1 protein turnover. In contrast to previous findings in COS7 and MRC5 cells (*Garvin et al., 2019*), in our RKO cell-based experiments, FBXW7 did not contribute to IRF1 degradation, as its ablation did not alter IRF1 protein concentrations. Nevertheless, FBXW7 cellular functionality and efficient targeting was demonstrated, since it was key to cMYC degradation, consistent with previous reports (*Welcker et al., 2004*; *Yada et al., 2004*). These findings expand our knowledge on the pathways that can be employed by different cells to balance IRF1 levels. Moreover, it underpins the notion that, as for cMYC, the factors mediating IRF1 turnover can be cell-context-specific. This notion of cell-type-specific regulation of protein degradation is further underpinned by the fact that we show in this manuscript that SPOP is dispensable for cMYC degradation in our cell models, whereas it has been reported to mediate cMYC turnover in prostate and breast cancers (*Geng et al., 2017*; *Luo et al., 2018*).

SPOP has been found to be differentially expressed in many cancers (*Cuneo and Mittag, 2019*; *Guo et al., 2016*; *Li et al., 2022*; *Patel et al., 2020*; *Song et al., 2020*; *Zhao et al., 2016*). Moreover, mutations accumulate in SPOP target recognition domains in endometrial and prostate cancers, altering their sensitivity to BET inhibitor therapy (*Dai et al., 2017*; *Janouskova et al., 2017*; *Zhang et al., 2017*). Interestingly, these mutations enhanced BRD protein degradation in endometrial cancers,

whereas it did the opposite in prostate cancers (*Dai et al., 2017*; *Janouskova et al., 2017*; *Zhang et al., 2017*). Thus, SPOP substrate targeting, its importance as a tumor-promoting or -repressing factor, and therapy sensitivity may be cancer- or cell-type specific. This exemplifies the cell context-dependent similarities between IRF1 and SPOP.

Several SPOP substrates important in a cancer context have been previously identified, such as BRD proteins, DRAK1 and LATS1 (*Pang et al., 2022*; *Wang et al., 2020*). Our data presented here adds IRF1 to this growing list of SPOP substrates. Since IRF1 levels are low in many cells in the absence of cytokine stimulation, it may have previously been missed as a SPOP interactor and substrate in unbiased proteomics efforts (*Dai et al., 2017*; *Janouskova et al., 2017*; *Zhang et al., 2017*). Thus, while the contributions of SPOP to cancer establishment or progression are likely through deregulation of a manifold of its substrates, our work suggests that IRF1 may be one of them. In line with our finding that SPOP could be critical for lowering the tumor-suppressive levels of IRF1, SPOP is overexpressed in most clear-cell renal cell carcinomas (ccRCC) (*Patel et al., 2020*; *Zhao et al., 2016*). The observation that IFNγ signaling and IRF1 expression are important biomarkers in ccRCC (*Kong et al., 2022*), addressing the importance of the SPOP-IRF1 axis in this cancer type may be of particular interest in future studies.

The complexity of IRF1 turnover regulation in different cell types is further illustrated by the fact that SPOP may influence IRF1 stability in myeloid cells in an opposite manner as we identified here in stromal cell types (*Tawaratsumida et al., 2022*). Recent work identified SPOP in myeloid cells as a key bridging molecule, indirectly repressing IRF1 phosphorylation, and its subsequent degradation. In line with this model, loss of *SPOP* in plasmacytoid dendritic cells and a mouse macrophage cell line decreased IRF1 (*Tawaratsumida et al., 2022*). This opposite SPOP phenotype and function from what we describe here could be explained by the fact that many non-immune cells do not express TLRs and their associated adapter proteins, which were shown to be required for SPOP complex formation and repression of IRF1 phosphorylation and degradation. It should be noted that the RKO cells used in this study have been reported to express TLRs (*Xu et al., 2020*). This suggests that additional cellular factors, such as recruited adaptors or differential dependence on specific Src kinases for signaling, may determine SPOP-dependent regulation in myeloid cells.

Importantly, IRF1 degradation in myeloid cells is seemingly dependent on active TLR-dependent immune signaling, whereas in the stromal, non-myeloid cell models used in this study, IRF1 degradation was independent of immune signaling. Several other studies have indicated that SPOP may inhibit TLR signaling through several different proposed means (*Hu et al., 2021*; *Jin et al., 2020*; *Li et al., 2020*), suggesting that SPOP may balance biological output from the TLR-IRF1 axis through opposing effects. Together, these possible contrary effects of SPOP on IRF1 abundance in different cell types should thus be considered when addressing the impact of SPOP in the context of future whole-organism studies.

During the revision of this manuscript, Gao et al. also reported SPOP as an important IRF1 degrader in endometrial cancer cells (*Gao et al., 2023*). They found that cancer-associated SPOP mutations result in diminished IRF1 degradation, and consequentially increased expression of the IRF1-driven immune-suppressive PD-L1. Most of the cell-mechanistic data of how SPOP targets IRF1 for ubiquitination and degradation are in agreement between both studies. However, a notable difference between both studies is that in our experiments, we showed that in RKO and HeLa cells, SPOP is localized in the nucleoplasm and nuclear speckles, whereas IRF1 is exclusively diffusely distributed in the nucleoplasm. Since proteasome inhibition increased only the diffuse nucleoplasmic IRF1 pool, we concluded that in our cell models SPOP most likely targets IRF1 in the nucleoplasm, and not in nuclear speckles. In contrast, Gao et al. found that IRF1 strongly associates with SPOP in nuclear speckles in KLE endometrial cancer cells (*Gao et al., 2023*). It remains unclear where this difference stems from. It could be that SPOP-IRF1 subcellular interaction sites differ in different cell types, or that technical differences – such as SPOP/IRF1 expression levels – influence subcellular interaction sites.

Previous work has indicated that SPOP recognizes nuclear targets with intrinsically disordered regions through multivalent interactions with S/T-rich degrons (*Usher et al., 2021*). In addition, its related paralog SPOP-like (SPOPL) has been proposed to regulate the oligomeric state of the SPOP E3 ligase complex (*Errington et al., 2012*). While our genetic screens indicate that SPOPL does likely not play an important role in IRF1 targeting, our data from this study are fully consistent with the reported substrate profile: IRF1 has IDRs and localizes almost exclusively to the nucleoplasm. Moreover, IRF1 is

likely targeted by SPOP through recognition of one or more S/T-rich degrons, although we have not determined the contribution of the four individual S/T-rich degrons to IRF1 degradation.

The S/T residues in these SBD are theoretical sites of phosphorylation, which as such could influence SPOP-IRF1 interaction. Indeed, previous work has indicated that SPOP preferentially binds some substrate S/T-rich degrons in their non-phosphorylated form, whereas in other substrates, phosphorylation of these sites seems a prerequisite for interaction and degradation (*Jiang et al., 2021*; *Ostertag et al., 2019*; *Zhuang et al., 2009*). Whether any of the identified IRF1 degrons are phosphorylated, and how this would affect its SPOP-dependent degradation, remains to be determined. IRF1 phosphorylation on S219 and S221 in the fourth degron was identified with low coverage in available IRF1 proteomics data (*Garvin et al., 2019*) and independently in a region that includes this site (*Lin and Hiscott, 1999*), suggesting that this particular identified SBD could be post-translationally modified. The same sites were identified to be phosphorylated in breast cancer cells by the immune-activating kinase IKKε (*Remoli et al., 2020*). Phosphorylation under these conditions increased IRF1 proteasomal degradation. Since IKKε is activated under physiological conditions during infection, this could suggest that in some cellular contexts, ongoing immune signaling could thus provide a regulatory circuit driving transcription of *IRF1*, as well as providing its subsequent signal for degradation of the encoded IRF1 protein by SPOP.

Three of the identified S/T-rich degrons in IRF1 are located in its IAD2 transactivation domain. This domain is key for heterodimerization with other transcriptional activators or repressors (*Antonczyk et al., 2019*). Therefore, an additional regulatory implication of our findings could be that IRF1 heterodimerization through its IAD2 domain could shield three of the S/T-rich degrons from recognition by SPOP. As such, SPOP could thus preferentially target the inactive nucleoplasmic IRF1 pool that is not engaged in transcription.

Taken together, our study identified and characterized SPOP as a cellular factor targeting IRF1 for proteasomal degradation. It extends the growing set of E3 ligases which collectively mediate proper intracellular IRF1 protein levels in various different cellular contexts. Our findings lay the foundation for future studies to address how transcriptionally inactive and active immune transcription factor pools are properly spatiotemporally targeted in diverse cell types.

# Materials and methods

## Vectors

The lentiviral human genome-wide sgRNA library consists of six sgRNAs per gene for a whole coding human genome and was described elsewhere (*Michlits et al., 2020*). Lentiviral vectors expressing sgRNA from a U6 promoter and eBFP2 or iRFP from a PGK promoter have been described elsewhere (*de Almeida et al., 2021*). sgRNA CDSs were cloned in pLentiv2-U6-PGK-iRFP670-P2A-Neo (*de Almeida et al., 2021*) to perform knockouts in RKO cell lines. The IRF1 and cMYC stability reporters (pLX-SFFV-mCherry-IRF1-P2A-eBFP2 and pLX-SFFV-mCherry-cMYC-P2A-eBFP2) were constructed by cloning the open reading frame (ORF) of human IRF1 or cMYC into a modified pLX303 vector (Addgene plasmid 25897). cDNAs encoding IRF1-SBD$_{mut}$, wtSPOP, and SPOP-MATH$_{mut}$ were purchased from Twist Bioscience. The IRF1-SBD$_{mut}$ degron mutant (S154A, S155A, T156A, S189A, S190A, T191A, S205A, T206A, S207A, S219A, T220A, S221A) and the SPOP-MATH$_{mut}$ binding mutant (Y87C, F102C, W131G) were cloned into a modified pLX303 vector. The plasmids and sgRNAs used in this study are listed in *Supplementary file 3* and *Supplementary file 4*.

## Cell culture

All experiments in this study have been reproduced at least twice in independent experiments. RKO-mCherry-IRF1-P2A-eBFP2 or RKO-mCherry-cMYC-P2A-eBFP2 cells were generated by transducing RKO-Dox-Cas9 cells (*de Almeida et al., 2021*) with SFFV-MYC-mCherry-IRF1-P2A-eBFP2 or SFFV-MYC-mCherry-cMYC-P2A-eBFP2 lentiviral vectors. Cas9 was induced with 100 ng/ml of doxycycline hyclate (Dox, Sigma-Aldrich, D9891) and mCherry-, eBFP2-, and GFP-positive single cells were sorted by FACS into 96-well plates using a FACSAria III cell sorter (BD Biosciences) to obtain single cell-derived clones. Cas9 genome editing and expression in the absence of Dox from the TRE3G promoter was tested in competitive proliferation assays. The cell lines and culture conditions used in this study are listed in *Supplementary file 5*. Cell lines used in this study were authenticated by STR analysis.

## Transfections

Transfections for analysis by western blot were performed by mixing DNA and PEI (Polysciences, 23966) in a 1:3 ratio (wt/wt) in DMEM (Sigma-Aldrich, D6429) without supplements. Transfection was performed using 500 ng of total DNA. The day before transfection, $1.5 \times 10^5$ HEK-293T cells were seeded in six-well clusters in fully supplemented DMEM media. Cells were harvested 36 hr after transfection, washed with ice-cold PBS, and stored at −80°C until further processing. For dual luciferase assays, HEK-293T cells in M24 well clusters were transfected with 170 ng ISRE-Fluc reporter, 30 ng pRL-TK (Rluc; Promega), 320 ng of wtIRF1 or 160 ng of IRF1-SBD$_{mut}$ expression plasmid, and 640 ng of wtSPOP or SPOP-MATH$_{mut}$ expression plasmid. Twenty-four hr after transfection, cells were lysed with Passive Lysis Buffer (Promega, E194A) and Firefly- and Renilla luciferase activities measured in a dual-luciferase assay using a Synergy H1 plate reader (BioTek). Substrates were prepared as previously reported (*Hampf and Gossen, 2006*).

## FACS-based CRISPR-Cas9 screens

Lentivirus-like particles were used to transduce RKO-mCherry-IRF1-P2A-eBFP2 or RKO-mCherry-cMYC-P2A-eBFP2 cells at an MOI of less than 0.2 TU/cell, and 500- to 1000-fold library representation. The percentage of library-positive cells was determined after 3 days of transduction by immunostaining of the Thy1.1 surface marker and subsequent flow cytometric analysis. RKO cells with integrated lentiviral vectors were selected with G418 (1 mg/ml, Sigma-Aldrich, A1720) for 7 days. After G418 selection, 120 million cells of unsorted reference sample corresponding to 1000-fold library representation were collected, and stored at −80°C until further processing. Cas9 genome editing was induced with Dox (100 ng/ml, Sigma-Aldrich, D9891) and after 3 days and 6 days, cells were sorted by FACS. Cells were harvested, washed with PBS, and stained with Fixable Viability Dye eFluor (1:1000, eBioscience, 65-0865-14) for 30 min. Subsequently, cells were washed three times with PBS and sorted in supplemented RPMI-1640 using the FACSAria III cell sorter operated by BD FACSDiva software (v8.0). RKO cells were gated for non-debris, singlets, negative for Viability Dye, GFP-positive, mCherry-positive, and BFP-positive. Two to 3% of cells with the lowest and 1–2% of cells with the highest mCherry or BFP signals were sorted into PBS. For both mCherry-IRF1 and mCherry-cMYC genome-wide screens, at least $1 \times 10^6$ (mCherry$^{low}$ and BFP$^{low}$) and $2 \times 10^6$ (mCherry$^{high}$ and BFP$^{high}$) cells were collected for each time point. Sorted samples were re-analyzed for purity, pelleted and stored at −80°C until further processing. The gating strategy for flow cytometric cell sorting is shown in *Figure 2—figure supplement 3* and *Figure 2—figure supplement 4*.

## Protein half-life determination

To estimate endogenous IRF1 and cMYC as well mCherry-IRF1 and mCherry-cMYC protein half-lives, RKO mCherry-IRF1 or mCherry-cMYC-expressing cells were treated with 30 µg/ml of CHX (Sigma-Aldrich, C1988). At indicated time points, total protein extracts were generated using RIPA buffer, analyzed by western blot, quantified, and normalized to ACTIN levels and to time point 0 as indicated. Single exponential decay curves were plotted using GraphPad Prism (v9), from which protein half-lives were calculated.

## Immunofluorescence confocal microscopy

200,000 HeLa cells/35 mm well were seeded onto coverslips (Marienfeld, 630-2190) and transfected using polyethylenimine (PEI) transfection reagent (Polysciences, 23966) in a 1:6 (wt/wt) DNA/PEI ratio in non-supplemented DMEM with lentiviral plasmids expressing MYC-tagged wtIRF1 and FLAG-tagged wtSPOP. 48 hr after transfection, cells were fixed with 4% paraformaldehyde (PFA) for 15 min. Similarly, 250,000 RKO cells stably expressing HA-tagged wtIRF1 and FLAG-tagged wtSPOP were seeded onto coverslips and after 48 hr stimulated with epoxomicin (10 µM, 4 hr) followed by fixation. Cells were permeabilized with 0.25% Triton X-100 in PBS for 5 min, followed by blocking of non-specific sites by incubation with 1% BSA for 30 min at RT. Coverslips were incubated for 1 hr at RT with primary anti-IRF1 antibody (Cell Signaling Technology, 1:100), anti-HA (Cell Signaling Technology, 1:1000), or anti-FLAG (M2, 1:500) antibody in 1% BSA, followed by incubation with anti-rabbit IgG Alexa Fluor 488 (Abcam, 1:800), anti-mouse Alexa Fluor 647 (Invitrogen, 1:1000), or anti-mouse IgG DyLight 550 (Invitrogen, 1:500) secondary antibodies, and incubation for 5 min with 0.4X Hoechst (Thermo Fisher Scientific, H3569) in PBS. The coverslips were mounted using ProLong Gold Antifade

Mountant (Invitrogen, P36934). Images were collected using a Zeiss LSM 980 confocal microscope at ×40 magnification.

## Subcellular fractionation

Subcellular fractionation was performed as previously described (*Suzuki et al., 2010*). In brief, 2 million cells were washed in 1 ml PBS and lysed in 500 µl ice-cold REAP buffer (0.1% NP-40 in 1x PBS) supplemented with 1 mM PMSF, 1X protease inhibitor cocktail (Sigma-Aldrich, 11836145001), and 25 U/ml benzonase. 240 µl of the lysates were collected as whole-cell fractions, the remaining lysate was centrifuged at 3000 × *g* for 60 s at 4°C. 240 µl of supernatants were collected as cytosolic fractions, after which pellets were washed with 500 µl of REAP buffer by centrifugation at 3000 × *g* for 60 s at 4°C, then resuspended in 240 µl of REAP buffer (nuclear fraction). All fractions were subsequently supplemented with 6x Laemmli sample buffer (62.5 mM Tris-HCl [pH 6.8], 5.8% glycerol, 2% SDS, and 1.7% β-mercaptoethanol), and boiled for 10 min. Equal volumes of fractions were loaded on a 10% SDS polyacrylamide gel.

## Western blot

Cells were lysed in RIPA lysis buffer (50 mM Tris HCl [pH 7.4], 150 mM NaCl, 1% NP-40, 0.5% sodium deoxycholate, 1 mM EDTA, 0.1% SDS, 0.1 mM PMSF, and 1X protease inhibitor cocktail). Cells were rotated for 20 min at 4°C and then centrifuged at 18,500 × *g* for 10 min at 4°C. Supernatants were transferred to new tubes and protein concentrations were determined by Pierce BCA Protein Assay Kit (Thermo Fisher Scientific, 23225). Between 20 and 50 µg of protein per sample was mixed with Laemmli sample buffer (62.5 mM Tris-HCl [pH 6.8], 5.8% glycerol, 2% SDS, and 1.7% β-mercaptoethanol), and boiled for 5 min. Proteins were loaded on SDS polyacrylamide gels, the percentage of which was adjusted based on the molecular weight of the proteins of interest. Proteins were blotted on PVDF or nitrocellulose membranes at 4°C for 1 hr at 300 mA in Towbin buffer (25 mM Tris pH 8.3, 192 mM glycine, and 20% ethanol). Membranes were blocked in 5% BSA in PBS-T for 1 hr at RT, and subsequently incubated with primary antibodies overnight at 4°C. The next day, the membranes were washed three times for 5 min each with PBS-T and incubated with HRP-coupled secondary antibodies for 1 hr at RT and imaged with the ChemiDoc Imaging System from Bio-Rad. Relative protein levels were quantified using Image Lab (Bio-Rad). Antibodies used in this study are listed in *Supplementary file 1*.

## Co-immunoprecipitation assays

Cells were lysed in 100 µl of Frackelton lysis buffer (10 mM Tris [pH 7.4], 50 mM NaCl, 30 mM $Na_4P_2O_7$, 50 mM NaF, 2 mM EDTA, 1% Triton X-100, 1 mM DTT, 1 mM PMSF, and 1X protease inhibitor cocktail). Cells were incubated on a rotating wheel at 4°C for 30 min and subsequently centrifuged at 20,000 × *g* at 4°C for 30 min. The supernatant was transferred to a new tube and 10 µl (10% of the lysate used for the immunoprecipitations [IPs]) was collected as input. 300 µg of lysates were incubated overnight at 4°C on a rotating wheel with an IgG Isotype Control (Cell Signaling Technology, 1:300), anti-Ollas (Novus, 1:100), or anti-IRF1 antibody (Cell Signaling Technology, 1:100). The next day, magnetic beads (Pierce Protein A/G Magnetic Beads, Thermo Fisher Scientific, 88803) used for anti-IRF1 antibody IPs, or Protein G Sepharose beads (Protein G Sepharose 4 Fast Flow, Sigma-Aldrich, GE17-0618-01) used for anti-Ollas antibody IPs, were blocked by rotation in 3% BSA in Frackelton Buffer for 1 hr at 4°C. 25 µl of beads were added to 300 µg of lysates and rotated for 2 hr at 4°C. Then, the beads were washed five times with 1 ml of RIPA buffer, supplemented with 300 mM NaCl. Proteins were eluted by boiling in 2X disruption buffer (2.1 M urea, 667 mM β-mercaptoethanol, and 1.4% SDS) for 5 min at 95°C.

## IPs for ubiquitination

Cells were lysed in 1 ml of RIPA lysis buffer (50 mM Tris-HCl [pH 7.4], 150 mM NaCl, 1% SDS, 0.5% sodium deoxycholate, 1% Triton X-100), supplemented with 40 mM *N*-ethylmaleimide, 40 mM iodoacetamide, 25 U/ml benzonase, 1 mM PMSF, and 1X protease inhibitor cocktail. Cells were incubated on a rotating wheel at 4°C for 30 min, and centrifuged at 20,000 × *g* at 4°C for 15 min. Supernatants were transferred to new tubes and 50 µl (20% of the lysates used for the IPs) were collected as input. 500 µg of lysates were incubated overnight at 4°C on a rotating wheel with an anti-IRF1 antibody (Cell

Signaling Technology, 1:100). The next day, magnetic beads (Pierce Protein A/G Magnetic Beads, Thermo Fisher Scientific, 88803) were blocked by rotation in 3% BSA in RIPA Buffer for 1 hr at 4°C. 25 µl of beads were added to 500 µg of lysates and rotated for 2 hr at 4°C. Subsequently, beads were washed five times with 1 ml of RIPA buffer, supplemented with 300 mM NaCl. Proteins were eluted by boiling in 2X disruption buffer (2.1 M urea, 667 mM β-mercaptoethanol, and 1.4% SDS) for 5 min at 95°C. Volumes loaded for the IP were adjusted based on the relative IRF1 amount in those samples, quantified by Image Lab.

### Lentivirus production and transduction

HEK-293T cells were transfected with DNA mixes containing lentiviral transfer plasmids, pCRV1-Gag-Pol (*Hatziioannou et al., 2004*) and pHCMV-VSV-G (*Yee et al., 1994*) using PEI (Polysciences, 23966) in a 1:3 µg DNA/PEI ratio in non-supplemented DMEM. Virus containing supernatants were clarified of cellular debris by filtration through a 0.45 µm filter. Virus-like particles were directly used after harvesting, or stored at –80°C. Target cells were transduced in the presence of 5 µg/ml polybrene (Sigma-Aldrich, TR1003G).

### Intracellular staining for flow cytometry

For staining of intracellular proteins, cells were collected, washed twice with PBS, and subsequently fixed with 2% PFA for 15 min at RT. After one wash with FACS buffer (PBS supplemented with 1% FCS), cells were permeabilized in ice-cold MeOH for 10 min and pelleted three times for 10 min at $900 \times g$. Subsequently, cells were incubated for 10 min at RT in FACS buffer to inhibit non-specific *antibody binding*. Cells were then incubated with primary antibodies, or left unstained for 1 hr at RT. Following two FACS buffer washes, cells were incubated with secondary antibodies for 15 min at 4°C., washed two times with FACS buffer and resuspended in FACS buffer for flow cytometric analysis on an LSRFortessa (BD Biosciences) operated by BD FACSDiva software (v8.0). FACS data were analyzed in FlowJo (v10.8).

### NGS library preparation

NGS libraries of sorted and unsorted control samples were processed as previously described (*de Almeida et al., 2021*). In brief, isolated genomic DNA was subjected to two-step PCR. The first PCR allowed the amplification of the integrated sgRNA cassettes, the second PCR introduced the Illumina adapters. Purified PCR products' size distribution and concentrations were measured using a fragment analyzer (Advanced Analytical Technologies). Equimolar ratios of the obtained libraries were pooled and sequenced on a HiSeq 2500 platform (Illumina). Primers used for library amplification are listed in *Supplementary file 7*.

### Analysis of pooled CRISPR screens

The analysis of the CRISPR-Cas9 screens was carried out as previously described (*de Almeida et al., 2021*). In brief, sgRNAs enriched in day 3 and day 6 post-Cas9 induction-sorted samples were compared against the matching unsorted control populations harvested on the same days using MAGeCK (*Li et al., 2014*). In addition, depletion of sgRNAs over time was calculated by comparing the unsorted populations to the initial day 0 population.

### Cell competition assays

Competitive cell fitness assays were performed as described previously (*Lee et al., 2012*). In brief, parental RKO cells harboring Dox-inducible Cas9 or the derived monoclonal RKO-mCherry-IRF1-P2A-BFP cell line were transduced with iRFP lentiviral sgRNA plasmids targeting either *AAVS1* or *SPOP*. The multiplicity of transduction was such that 30–60% of cells were iRFP-positive before the start of fitness measurements. Gene editing was induced with Dox and the percentage of iRFP-positive cells monitored for 20 days by flow cytometry at the indicated days. The relative fraction of sg*SPOP*-positive cells was normalized to sg*AAVS1* of the same cell line on day 0. Data were analyzed by two-way ANOVA.

### VSV infections and titration

RKO-mCherry-IRF1-P2A-eBFP2 cells harboring *STAT1* and *IRF3* sgRNA expression vectors were additionally transduced with vectors expressing either sg*AAVS1* or sg*SPOP*. Subsequently, sgRNA-positive

cells were collected by FACS. After Cas9 induction by Dox, cells were seeded in M24 clusters, and infected with VSV Indiana at an MOI of 0.0005 PFU/cell as previously described (*Mata et al., 2011*). After 26 hr, supernatants were collected, clarified, and viral titers determined by plaque assays as previously described (*Matrosovich et al., 2006*).

## RNA isolation, cDNA synthesis, and qPCR

Total RNA was extracted from RKO-Dox-Cas9-P2A-GFP cells harboring either *AAVS1/CCR5*, *AAVS1/IRF1*, *SPOP/CCR5*, *SPOP/IRF1*-targeting sgRNAs. $1 \times 10^6$ cells were lysed using Trizol reagent (Thermo Fisher Scientific, 5596-018) and treated with Turbo DNase (Thermo Fisher Scientific, 10792877). cDNA was prepared using random hexamer primer and RevertAid Reverse Transcriptase (Thermo Fisher Scientific, EP0441). Real-time PCR experiments were run on a Mastercycler (Bio-Rad), using SYBR Green (Thermo Fisher Scientific). Primers for qPCR are listed in *Supplementary file 6*.

## Materials availability statement

All data generated or analyzed during this study are included in the manuscript and supporting files.

## Acknowledgements

We thank Kitti Csalyi, Johanna Stranner, and Thomas Sauer at the Max Perutz Labs BioOptics FACS Facility for expert support, the Vienna Biocenter Core Facilities (VBCF) for Next Generation Sequencing analysis, and Aleksej Drino for help with plotting of NGS data. The MEFs used in this study were a kind gift from Manuela Baccarini. We are grateful to the 'Signaling Mechanisms in Cellular Homeostasis' doctoral program community, Manuela Baccarini, Pavel Kovarik, and their labs for their technical expertise and help. Funding sources: This work was funded by Stand-Alone grants (P30231-B, P30415-B), Special Research Grant (SFB grant F79), and Doctoral School grant (DK grant W1261) from the Austrian Science Fund (FWF) to GAV, a Starting Grant from the European Research Council (ERC-StG-336860) to JZ, the Austrian Science Fund (SFB grant F4710) to JZ, and Stand-Alone grant (P25186-B22), Special Research Grant (SFB grant F6103), and Doctoral School grant (DK grant W1261) from the Austrian Science Fund to TD. MV, VB, and MdeA are recipients of a DOC fellowship of the Austrian Academy of Sciences. Research at the IMP is supported by Boehringer Ingelheim and the Austrian Research Promotion Agency (Headquarter grant FFG-852936).

## Additional information

### Competing interests

Johannes Zuber: JZ is a founder, shareholder and scientific advisor of Quantro Therapeutics GmbH. The Zuber lab receives research support and funding from Boehringer Ingelheim. The other authors declare that no competing interests exist.

### Funding

| Funder | Grant reference number | Author |
|---|---|---|
| Austrian Science Fund | Stand-Alone grant (P30231-B) | Gijs A Versteeg |
| Austrian Science Fund | Stand-Alone grant (P30415-B) | Gijs A Versteeg |
| Austrian Science Fund | Special Research grant (SFB grant F79) | Gijs A Versteeg |
| Austrian Science Fund | Doctoral School grant (DK grant W1261) | Thomas Decker Gijs A Versteeg |
| European Research Council | ERC-StG-336860 | Johannes Zuber |
| Austrian Science Fund | SFB grant F4710 | Johannes Zuber |

| Funder | Grant reference number | Author |
| --- | --- | --- |
| Austrian Science Fund | Stand-Alone grant (P25186-B22) | Thomas Decker |
| Austrian Science Fund | Special Research Grant (SFB grant F6103) | Thomas Decker |
| Austrian Academy of Sciences | DOC fellowship | Milica Vunjak Valentina Budroni Melanie de Almeida |

The funders had no role in study design, data collection and interpretation, or the decision to submit the work for publication.

## Author contributions

Irene Schwartz, Milica Vunjak, Formal analysis, Investigation, Methodology, Writing - original draft, Writing - review and editing; Valentina Budroni, Formal analysis, Investigation, Methodology, Writing - review and editing; Adriana Cantoran García, Marialaura Mastrovito, Formal analysis, Investigation, Methodology; Adrian Soderholm, Kathrin Hacker, Formal analysis, Investigation; Matthias Hinterndorfer, Melanie de Almeida, Julian Jude, Resources, Methodology; Jingkui Wang, Kimon Froussios, Formal analysis; Thomas Decker, Johannes Zuber, Resources, Investigation; Gijs A Versteeg, Conceptualization, Supervision, Funding acquisition, Investigation, Writing - original draft, Project administration, Writing - review and editing

## Author ORCIDs

Irene Schwartz ⓘ http://orcid.org/0009-0006-9499-1555
Milica Vunjak ⓘ http://orcid.org/0000-0001-6330-363X
Valentina Budroni ⓘ http://orcid.org/0000-0002-6606-2031
Matthias Hinterndorfer ⓘ http://orcid.org/0000-0003-2435-4690
Jingkui Wang ⓘ http://orcid.org/0000-0002-0127-501X
Kimon Froussios ⓘ http://orcid.org/0000-0003-2812-0525
Julian Jude ⓘ http://orcid.org/0000-0002-9091-9867
Thomas Decker ⓘ http://orcid.org/0000-0001-9683-0620
Johannes Zuber ⓘ https://orcid.org/0000-0001-8810-6835
Gijs A Versteeg ⓘ http://orcid.org/0000-0002-6150-2165

## Decision letter and Author response

Decision letter https://doi.org/10.7554/eLife.89951.sa1
Author response https://doi.org/10.7554/eLife.89951.sa2

# Additional files

## Supplementary files

• Supplementary file 1. Antibodies used in this study.

• Supplementary file 2. Reagents used in this study.

• Supplementary file 3. Plasmids used in this study.

• Supplementary file 4. sgRNA sequences used in this study.

• Supplementary file 5. Cell lines used in this study.

• Supplementary file 6. qPCR primers used in this study.

• Supplementary file 7. Next-generation sequencing (NGS) library PCR primers used in this study.

• Supplementary file 8. Results from CRISPR-Cas9 genetic screens of mCherry-IRF1 and mCherry-cMYC.

• MDAR checklist

## Data availability

All data generated or analysed during this study are included in the manuscript and supporting files.

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
