## [Editor Report]

IRF1 is a key transcription factor with important roles during pathogenic infection and in some cancers. The regulation of IRF1 protein abundance is critical, but the underlying mechanisms are not known. Using a CRISPR-based knockout screen, the authors identify SPOP as an E3 ligase that binds IRF1 and enforces its degradation via the proteasome. The study represents an important advance backed by solid evidence that may have implications in immunology, virology, and cancer fields.

---

## [Decision Letter]

**Decision letter after peer review:**

[Editors’ note: the authors submitted for reconsideration following the decision after peer review. What follows is the decision letter after the first round of review.]

Thank you for submitting the paper "SPOP targets the immune transcription factor IRF1 for proteasomal degradation" for consideration by *eLife*. Your article has been reviewed by 3 peer reviewers, one of whom is a member of our Board of Reviewing Editors, and the evaluation has been overseen by a Senior Editor. The reviewers have opted to remain anonymous.

Comments to the Authors:

We are sorry to say that, after consultation with the reviewers, we have decided that this work in its current form will not be considered for publication by *eLife*. Should the authors wish to address reviewers' comments, we would be willing to consider a substantially revised manuscript. In particular, a revised study should include experiments to (1) provide more robust evidence of SPOP-mediated ubiquitination of IRF1, (2) provide a deeper understanding of the localization of SPOP-mediated regulation of IRF1 (nuclear, cytoplasmic, during translocation), and (3) assess the biological relevance of SPOP/IRF1 axis in other cells (per reviewers' suggestions below).

*Reviewer #1 (Recommendations for the authors):*

The authors sought to address an important question of how IRF1 is turned over, as aberrant IRF1 expression is linked to several immunological disorders. The authors devised a clever fluorescent protein-based IRF1 degradation assay and used this system to perform a well-controlled genome-wide CRISPR screen for genes that regulated IRF1 expression. They identified SPOP as a hit and then carried out a series of experiments (genetic reconstitution, binding assays, localization) to validate SPOP as an E3 ligase that mediates the degradation of IRF1. In functional assays, they linked SPOP1-dependent IRF1 regulation to predicted changes in specific downstream IRF1 target genes, and they demonstrated modest effects of this pathway on cell fitness.

Strengths

Powerful CRISPR screens to identify SPOP as a regulator of IRF1 protein abundance;

Rigorous validation of hit, including binding assays, colocalization, and judicious use of both SPOP and IRF1 mutants as control.

Weaknesses

Current evidence for SPOP directly ubiquitinating IRF1 is incomplete and lacks complementing orthogonal assays;

The significance of the SPOP/IRF1 axis in a biologically relevant model system (infection, cancer, etc) is lacking.

1. The key/new mechanism here is that SPOP directly ubiquitinates IRF1 and enforces its degradation. There is only one Ub assay directly showing this, thus the strength of the evidence is incomplete. The authors need to provide additional data supporting this mechanism. Orthogonal assays (e.g. Ub assays in cells vs in vitro ubiquitination assays using recombinant protein) showing the same result are critical to pin this mechanism down. The use of SPOP mutants that cannot ubiquitinate IRF1 also needs to be included as controls.

2. The broader significance of the paper is uncertain in the absence of a more compelling role for the SPOP/IRF1 axis in a biologically relevant setting. The authors claim in the discussion that viral assays would be difficult due to the overlap of IRF1 and ISGF3 target genes. This logic is confusing. There are ample examples in the literature of IRF1 expression conferring antiviral effects to cells, especially in cells that can't signal through IFN (STAT1-deficient). Co-expression of SPOP with IRF1 should reduce IRF1 levels and create a more permissive antiviral state. Alternatively, a more robust cancer-related model could help assign a clearer biological role for SPOP/IRF1.

3. The authors should include a new discussion on how their work relates to a recent similar study (PMID: 36481790).

*Reviewer #2 (Recommendations for the authors):*

Vunjak and colleagues report the results of a genome-wide CRISPR screen designed to identify genes controlling the post-translational fate of IRF1. IRF1 was the first interferon regulatory factor to be identified, and its activity as a transcription factor has been associated with both successful immune control of intracellular viral and bacterial infections, as well as regulation of cell cycle progression and tumor suppression. Unlike IRFs such as IRF3, which are expressed constitutively and activated via phosphorylation, IRF1 activity is determined primarily by its protein abundance which is regulated transcriptionally, translationally (at the level of protein synthesis), and by the rate of protein degradation. A better understanding of these processes would be important for scientists with interests in innate immune host responses to pathogens as well as the role of IRF1 in protection against cancer.

Vunjak et al. focused on host factors controlling IRF1 protein degradation in RKO cells, a poorly differentiated human colonic carcinoma cell line, after confirming that the abundance of IRF1 protein in these cells was stabilized by inhibition of the proteasome when cells were treated with cycloheximide to block new protein synthesis. The authors constructed a cell line with stable expression of an mCherry-IRF1 reporter protein and conditional expression of Cas9, which they then transduced with lentiviruses expressing a genome-wide guide RNA (sgRNA) library. They used flow cytometry to select cells with enhanced mCherry-ORF1 expression, and high throughput sequencing of the selected cells to identify SPOP as a leading candidate for regulation of IRF1 protein abundance, putatively controlling its degradation. The authors went on to show in a series of validation experiments that targeted depletion of SPOP (a 374aa adapter protein that is a component of a cullin-RING E3 ubiquitin ligase complex involved in ubiquitination of a number of other cellular proteins) enhances endogenous IRF1 protein abundance, as anticipated from the screen. They show SPOP forms a complex with IRF1 through interactions with one or more predicted SPOP-interaction domains ('degrons') in IRF1, and provide data suggestive of SPOP-regulated IRF1 ubiquitination.

These experiments were generally well-controlled and carried out in a thoughtful manner, and collectively they provide solid evidence that SPOP regulates IRF1 abundance in RKO cells, most likely by mediating its ubiquitination and subsequent proteasome-dependent degradation. These results differ significantly from those of Tawaratsumida et al. (PMID 35857476), who reported earlier this year that the deletion of SPOP decreased, rather than increased, the abundance of IRF1 (and other IRFs) by increasing the activity of the Src kinase family member LYN and subsequent LYN-directed IRF degradation. However, they are consistent with a recent report by Gao et al. (PMID: 36481790), who also demonstrated that SPOP interacts with IRF1 and regulates IRF1 abundance via ubiquitination and proteasome-mediated degradation in endometrial carcinoma cells, thereby suppressing IRF1-mediated expression of the programmed cell death 1 ligand, PD-L1.

While providing valuable insight, there are several important limitations to the Vunjak study. As in all such CRISPR screens, genes that are essential for cell proliferation/survival and also involved in IRF1 regulation might be missed due to the global effects of gene knockout. More specific to the authors' screen, however, the half-life of the mCherry-IRF1 reporter protein was approximately twice that of endogenous IRF1 in cycloheximide-treated RKO cells, suggesting that the reporter protein might not be subject to an additional, SPOP-independent degradation pathway acting only on the endogenous protein. Another major limitation of the study is that the fate of IRF1 was investigated only in a single cancer cell line. IRF1 is recognized to have important tumor suppressor activity, and the control of its expression could differ from the norm in cells that have undergone a neoplastic transformation. Thus, while both the Gao report shows SPOP regulates IRF1 abundance directly in other cancer cell types, it would be important to know whether it similarly controls IRF1 abundance in non-neoplastic stromal cells, or in immune cells in the presence or absence of IRF1 activation by pathogen invasion.

Another uncertainty is the cellular compartment in which IRF1 degradation is regulated – is it regulated in the cytoplasm, or in the nucleus where most SPOP is expressed, or could it be regulated at the level of nuclear translocation? The latter possibility is particularly intriguing, as Vunjak's CRISPR screen identified multiple components of the nuclear pore complex in addition to SPOP. Also unknown is the impact of IRF1 phosphorylation on its degradation, either at the level of SPOP binding to IRF1, ubiquitination, or nuclear translocation of IRF1. While the authors note these questions, the manuscript leaves them unanswered, to be resolved hopefully in future studies.

RKO is a cancer cell line – is IRF1 degradation in these cells representative of IRF1 degradation pathways in normal cells? For example, FBXW7 appears to be not important for IRF1 regulation in these cells but has been reported to be so in other cell types.

Line 135-236 – The authors should consider that IRF1 abundance is regulated not only transcriptionally, but also translationally, as IRF1 synthesis is subject to inhibition by PKR independently of transcription (Feng et al., PMID 28967880).

Line 172-175 – The fact that the t_1/2_ of the mCherry-IRF1 product is twice as long as that of endogenous IRF1 suggests the possibility that the endogenous degradation pathway may be only partly represented/active in the reporter cell line.

Line 231 – (a) The authors state that SPOP was "The strongest candidate enriched at both three and six days after Cas9 induction" – however, SPOP is not shown in Figure S2E (the day 3 panel), and it is not the most significantly enriched in Figure 2D. (b) Were each of the 6 sgRNAs targeting SPOP enriched at day 3 and day 7?

Line 241-244 – "FBXW7, MDM2, [and] STUB1/CHIP … are unlikely to play a significant role in IRF1 degradation" – or, could these proteins be required for cell survival? The authors should note the limitations of this CRISPR screen.

Figure 3A – BFP labeling appears to be increased in the sgSPOP transduced cells. Was the change in MFI from sgAAVS1 significant?

Figure S3E (and other bar graphs) – showing symbols for the individual values reflected in the means shown by the bars would be informative and enhance the presentation of the authors' data.

Line 260-62 – Is the increase in IRF1 protein primarily cytoplasmic, or nuclear?

Lines 298-99, 318 – The data do not show that "the four predicted degrons collectively are required for SPOP-dependent IRF1 degradation" since all four degrons were mutated. It could be that only one of the degrons is functional in the wild-type protein and that the loss of degradation requires mutation of only that specific degron. The authors recognize this limitation in the Discussion on lines 472-473, but ought to change the text in Results to more accurately reflect the data. The Abstract should also be modified to reflect this more accurate interpretation of the data.

Figure 4E-H – Were the wt and mutated IRF1 expression products tagged with Myc? It is otherwise not clear why blotting was with antibody to Myc-IRF1 (minor detail).

Figure 4J, I – How reproducible were these IRF1 ubiquitylation experiments? There are no error bars in Figure 4I suggesting that the experiment may have been done only once.

The authors should cite the recent publication by Gao et al. (PMID: 36481790) in the Discussion, and compare the results of Gao et al., with those reported here.

*Reviewer #3 (Recommendations for the authors):*

In this study, Vunjak and colleagues show that the Speckle Type BTB/POZ Protein (SPOP) is responsible for the ubiquitination and turnover of Interferon Regulatory Factor 1 (IRF1). Using a CRISPR genetic screening approach, the authors identified several factors, including the E3 ligase substrate adaptor SPOP as regulators of IRF1 protein levels. The screening approach used good controls (such as Myc instead of IRF1) and showed significant results. Further, the authors identified potential SPOP-binding motifs in IRF1 and showed that they (or at least one of them) are important for the recognition and turnover via SPOP; they used flow cytometry, co-immunoprecipitation, western immunoblotting, and dual luciferase assays. Finally, to determine the response of IRF1 transcriptional activity to the presence and absence of SPOP, the authors performed a qPCR analysis of IRF1-driven transcripts, which revealed a significant increase in SPOP knockout cells.

The manuscript is well written. The authors find another substrate of SPOP. One open question is how significant this E3 ligase is for the turnover of IRF1. The difference in ubiquitination in WT and SPOP-binding motif-devoid IRF1 is modest. The differences in transcriptional activity, however, are large. A set of additional experiments are necessary to understand these counterintuitive effects. The authors identify 4 SPOP-binding motifs in IRF1 and suggest that they work synergistically in the recognition of IRF1 by SPOP and its subsequent degradation. However, without testing the contribution of individual SPOP-binding motifs, it is impossible to say whether they all contribute to SPOP turnover.

1. The current version of Figure 1E is weak in supporting the idea that IRF1 degradation is independent of autophagy/autolysosomal degradation. Additional controls including time-dependent CHX treatment (like in Figure 1A) and -IFNγ in the presence of Bafilomycin will help to answer this question clearly.

2. Is IRF1 ubiquitinated by SPOP also in the absence of inflammation or is this an inflammation-specific response?

3. There are no error bars in some histogram plots (Figure 3E, Figure 3F, Figure 4J, Figure 5C). How many biological and technical replicates did the authors perform?

4. In Figure 4D, it seems that both IRF1 and SPOP localize in the nucleus but the resolution of the microscopy image is not sufficient to clearly understand their subcellular distribution. SPOP is often localized to nuclear bodies such as nuclear speckles. Micrographs with better resolution and concentration on the nucleus are required.

5. The decrease in the ubiquitylation level of IRF1 with mutated SB motifs compared to IRF1 WT is modest (Figure 4I). SPOP seems to be responsible for the turnover of only a small fraction of IRF1. The authors found other E3 ligase hits in their genetic screen, e.g. HUWE1 (a strong candidate hit in Figure S2F). Have they considered testing whether there are other E3 ligases involved in the turnover of IRF1?

6. In Figure 5A and 5B, to be able to unambivalently identify whether tested transcripts are only driven by the transcriptional activity of IRF1 but not by other IRF family members or other transcription factors when treated with IFNγ, a control condition like sgIRF1 cells are necessary.

7. Experiments in Figure 5C and Figure 5D show a comparison of +/- WT SPOP expression. In Figure 5C, the ratio is about 10fold, but in Figure 5D around 100 fold. What results in this large deviation?

8. The authors observe little to no protein turnover effect of SPOP on cMYC. However, it has been previously shown that SPOP recognizes cMYC as a ubiquitination substrate (PMID: 28414305, PMID: 30002443). Could the authors please discuss the potential reasons for this discrepancy?

9. The authors mentioned a recent study (Tawaratsumida et al., 2022) that shows that deletion of SPOP results in LYN-dependent (Src kinase family member LYN) degradation of IRF1, and inhibition of its transcriptional activity. The discussion (in lines 452-456) suggests that the differences may arise from the expression of TLRs and associated adapters exclusively in immune cells. However, colon/colorectal cancer cells including RKO cells are known to express most TLRs (e.g. PMID: 33244454). It thus seems that TLR expression is not the critical difference that causes the differing response. Could the authors please discuss what other biological reasons could potentially cause this different response?

10. The statement "…the four predicted degrons collectively are required for SPOP-dependent IRF1 degradation." (lines 298-299) cannot be made without testing the response of the SPOP degrons individually, as well as in binary and ternary combinations. It is possible that only one of the motifs is responsible for the observed effect.

---

## [Author Response]

[Editors’ note: the authors resubmitted a revised version of the paper for consideration. What follows is the authors’ response to the first round of review.]

Comments to the Authors:We are sorry to say that, after consultation with the reviewers, we have decided that this work in its current form will not be considered for publication by eLife. Should the authors wish to address reviewers' comments, we would be willing to consider a substantially revised manuscript. In particular, a revised study should include experiments to1) Provide more robust evidence of SPOP-mediated ubiquitination of IRF1.

New cell-based ubiquitination data were added as figures 4I and 4J. These data address IRF1 ubiquitination by SPOP, using IRF1 mutants that lack SPOP-binding domains, and SPOP mutants lacking an intact substrate recognition domain. These new data show robust wtIRF1 ubiquitination by wtSPOP, which is severely reduced if either the E3 or substrate were mutated in their interaction surfaces. These new data provide evidence for robust IRF1 poly-ubiquitination, supported by quantification of replicates and statistical analysis.

2) Provide a deeper understanding of the localization of SPOP-mediated regulation of IRF1 (nuclear, cytoplasmic, during translocation), and

New data from immunofluorescence confocal microscopy and subcellular fractionation experiments were added to the revised manuscript as Figures 4A-B and Figure 4—figure supplement 1A-C. These data show that SPOP localizes in our cell models diffusely in the nucleoplasm and nuclear speckles, whereas IRF1 is exclusively diffusely nucleoplasmic. Additional data show that IRF1 only accumulated as a diffusely nucleoplasmic pool upon proteasome inhibition. In line with these findings, IRF1 and SPOP subcellular fractionation indicated that SPOP targets IRF1 in the nucleus, and that IRF1 is degraded in the nucleus. The fact that IRF1 was undetectable in the cytosol, indicates that IRF1 is likely rapidly imported into the nucleus upon synthesis, and that this is the predominant location where SPOP-dependent IRF1 degradation takes place. These results are in line with the findings from our genetic screen, which showed that knock-out of various nucleopore components resulted in increased cellular IRF1 concentrations (Figure 2E).

3) Assess the biological relevance of SPOP/IRF1 axis in other cells (per reviewers' suggestions below).

New data in Figure 3G demonstrate SPOP-dependent IRF1 degradation in mouse embryonic fibroblasts. These data show that our previous data from human cancer cell lines, are likewise relevant across species in non-cancer-derived cells from mice.

Reviewer #1 (Recommendations for the authors):The authors sought to address an important question of how IRF1 is turned over, as aberrant IRF1 expression is linked to several immunological disorders. The authors devised a clever fluorescent protein-based IRF1 degradation assay and used this system to perform a well-controlled genome-wide CRISPR screen for genes that regulated IRF1 expression. They identified SPOP as a hit and then carried out a series of experiments (genetic reconstitution, binding assays, localization) to validate SPOP as an E3 ligase that mediates the degradation of IRF1. In functional assays, they linked SPOP1-dependent IRF1 regulation to predicted changes in specific downstream IRF1 target genes, and they demonstrated modest effects of this pathway on cell fitness.StrengthsPowerful CRISPR screens to identify SPOP as a regulator of IRF1 protein abundance;Rigorous validation of hit, including binding assays, colocalization, and judicious use of both SPOP and IRF1 mutants as control.WeaknessesCurrent evidence for SPOP directly ubiquitinating IRF1 is incomplete and lacks complementing orthogonal assays;The significance of the SPOP/IRF1 axis in a biologically relevant model system (infection, cancer, etc) is lacking.1. The key/new mechanism here is that SPOP directly ubiquitinates IRF1 and enforces its degradation. There is only one Ub assay directly showing this, thus the strength of the evidence is incomplete. The authors need to provide additional data supporting this mechanism. Orthogonal assays (e.g. Ub assays in cells vs in vitro ubiquitination assays using recombinant protein) showing the same result are critical to pin this mechanism down. The use of SPOP mutants that cannot ubiquitinate IRF1 also needs to be included as controls.

New data in Figures 4I+J address IRF1 ubiquitination by SPOP, using IRF1 mutants that lack SPOP-binding domains, and SPOP mutants lacking an intact substrate recognition domain. These new data show robust wtIRF1 ubiquitination by wtSPOP, which was severely reduced if either the E3 or substrate were mutated in their interaction surfaces. These new data provide evidence for robust IRF1 poly-ubiquitination, supported by quantification of replicates and statistical analysis.

2. The broader significance of the paper is uncertain in the absence of a more compelling role for the SPOP/IRF1 axis in a biologically relevant setting. The authors claim in the discussion that viral assays would be difficult due to the overlap of IRF1 and ISGF3 target genes. This logic is confusing. There are ample examples in the literature of IRF1 expression conferring antiviral effects to cells, especially in cells that can't signal through IFN (STAT1-deficient). Co-expression of SPOP with IRF1 should reduce IRF1 levels and create a more permissive antiviral state. Alternatively, a more robust cancer-related model could help assign a clearer biological role for SPOP/IRF1.

New data in Figure 5F + Figure 5—figure supplement 1C address the relevance of SPOP for exogenous IRF1-driven antiviral responses in *STAT1*-targeted cells. The data show that exogenous IRF1 expression reduced VSV progeny virus production by ~10 fold, and that this was further reduced by approximately another 10-fold in the absence of SPOP. These results underpin the biological significance of SPOP in controlling the IRF1-driven antiviral state.

3. The authors should include a new discussion on how their work relates to a recent similar study (PMID: 36481790).

This reference was added to the Discussion section of the revised manuscript, as well as additional text comparing our results to the work of Gao *et al.*

Reviewer #2 (Recommendations for the authors):Vunjak and colleagues report the results of a genome-wide CRISPR screen designed to identify genes controlling the post-translational fate of IRF1. IRF1 was the first interferon regulatory factor to be identified, and its activity as a transcription factor has been associated with both successful immune control of intracellular viral and bacterial infections, as well as regulation of cell cycle progression and tumor suppression. Unlike IRFs such as IRF3, which are expressed constitutively and activated via phosphorylation, IRF1 activity is determined primarily by its protein abundance which is regulated transcriptionally, translationally (at the level of protein synthesis), and by the rate of protein degradation. A better understanding of these processes would be important for scientists with interests in innate immune host responses to pathogens as well as the role of IRF1 in protection against cancer.[…]RKO is a cancer cell line – is IRF1 degradation in these cells representative of IRF1 degradation pathways in normal cells? For example, FBXW7 appears to be not important for IRF1 regulation in these cells but has been reported to be so in other cell types.

New data in Figure 3G demonstrate SPOP-dependent IRF1 degradation in mouse embryonic fibroblasts. These data show that our previous data from human cancer cell lines are likewise relevant across species in non-cancer-derived cells from mice.

Line 135-236 – The authors should consider that IRF1 abundance is regulated not only transcriptionally, but also translationally, as IRF1 synthesis is subject to inhibition by PKR independently of transcription (Feng et al., PMID 28967880).

Additional text and the indicated reference was added to the Introduction section to reflect that endogenous IRF1 protein concentrations are also controlled at the translational level. It should be noted though that our screens and most other experiments were performed in the absence of infection. We reason that it is thus unlikely that our experiments were influenced by PKR-dependent translational shut-down, as viral RNA needed for PKR activation was absent.

Line 172-175 – The fact that the t_1/2_ of the mCherry-IRF1 product is twice as long as that of endogenous IRF1 suggests the possibility that the endogenous degradation pathway may be only partly represented/active in the reporter cell line.

Additional text was added to the Introduction section to reflect this point.

Line 231 – (a) The authors state that SPOP was "The strongest candidate enriched at both three and six days after Cas9 induction" – however, SPOP is not shown in Figure S2E (the day 3 panel), and it is not the most significantly enriched in Figure 2D. (b) Were each of the 6 sgRNAs targeting SPOP enriched at day 3 and day 7?

SPOP was marked in green in the original manuscript, but it was by mistake not labeled with its name. This has been rectified in the revised figure panel. The accompanying text was meant to convey that SPOP was the only IRF1-specific candidate enriched at both screening days, thereby making it the strongest IRF1-specific candidate identified by the screens. The relevant text in the Results section was updated to better convey this intended notion.

Line 241-244 – "FBXW7, MDM2, [and] STUB1/CHIP … are unlikely to play a significant role in IRF1 degradation" – or, could these proteins be required for cell survival? The authors should note the limitations of this CRISPR screen.

FBXW7 is a strong hit in the cMYC screen, but does not score at all in the IRF1 screen. As discussed in the manuscript, this is a reasonably strong indication that the experimental system works, but that FBXW7 targeting does not affect IRF1. For the other two factors, we cannot say this with as much confidence since they do not score in either screen. The text has been adapted to reflect that notion of uncertainty.

As pointed out in the manuscript, and further substantiated by our previous screens using the same screening approach (de Almeida et al. *Science* 2023, and Scinicariello et al. *eLife* 2023), the used screening setup allows for identification of cell-essential genes at the earlier time points post Cas9 induction with Dox (3 days post-Dox). However, by day 6, these cells will be lost. This notion is underpinned by the fact that we readily identify proteasome components, and other highly cell-essential degradation mediators like AKIRIN2 in the screens presented in this manuscript, and the two publications listed above. Moreover, the used screening cell lines containing an sgRNA targeting the highly cell-essential gene *RRM1*, were only lost upon addition of Dox to induce Cas9, but not without it (Figure 1—figure supplement 1G-H), further substantiating the great care taken to enable identification of cell-essential genes in our screens, without the risk of out-selection of cells in which essential genes were targeted prior to the screening procedure.

Figure 3A – BFP labeling appears to be increased in the sgSPOP transduced cells. Was the change in MFI from sgAAVS1 significant?

The BFP MFI did not significantly change in sg*SPOP* cells in this panel, nor did it in other experiments in which dual-color stability reporters were used.

Figure S3E (and other bar graphs) – showing symbols for the individual values reflected in the means shown by the bars would be informative and enhance the presentation of the authors' data.

Based on this comment, data in Figures 3H, 5F, and Figure 5—figure supplement 1B is now presented as individual data points. However, we opted for not displaying the individual data points in other figure panels as it made them overly complex in our opinion, and less clear for that reason.

Line 260-62 – Is the increase in IRF1 protein primarily cytoplasmic, or nuclear?

The revised manuscript includes new data on this (Figure 4A-B, and Figure 4—figure supplement 1A-C). We did not detect cytosolic IRF1. However, we did measure an increase in the nucleoplasmic IRF1 pool upon *SPOP* knock-out, indicating that this likely reflects the targeted IRF1 protein population.

Lines 298-99, 318 – The data do not show that "the four predicted degrons collectively are required for SPOP-dependent IRF1 degradation" since all four degrons were mutated. It could be that only one of the degrons is functional in the wild-type protein and that the loss of degradation requires mutation of only that specific degron. The authors recognize this limitation in the Discussion on lines 472-473, but ought to change the text in Results to more accurately reflect the data. The Abstract should also be modified to reflect this more accurate interpretation of the data.

We agree, and changed the text in the Abstract and Manuscript to reflect this.

Figure 4E-H – Were the wt and mutated IRF1 expression products tagged with Myc? It is otherwise not clear why blotting was with antibody to Myc-IRF1 (minor detail).

Yes, wtIRF1 and IRF1-SBM were tagged with a N-terminal myc-tag and detected with an anti-myc antibody.

Figure 4J, I – How reproducible were these IRF1 ubiquitylation experiments? There are no error bars in Figure 4I suggesting that the experiment may have been done only once.

New data in Figure 4I-J address IRF1 ubiquitination by SPOP, using IRF1 mutants that lack SPOP-binding domains, and SPOP mutants lacking an intact substrate recognition domain. These new data show robust wtIRF1 ubiquitination by wtSPOP, which is severely reduced if either the E3 or substrate were mutated in their interaction surfaces. These new data provide evidence for robust IRF1 poly-ubiquitination, supported by quantification of replicates and statistical analysis in Figure 4J.

The authors should cite the recent publication by Gao et al. (PMID: 36481790) in the Discussion, and compare the results of Gao et al., with those reported here.

This reference was added to the Discussion section of the revised manuscript, as well as additional text comparing our results to the work of Gao *et al.*

Reviewer #3 (Recommendations for the authors):In this study, Vunjak and colleagues show that the Speckle Type BTB/POZ Protein (SPOP) is responsible for the ubiquitination and turnover of Interferon Regulatory Factor 1 (IRF1). Using a CRISPR genetic screening approach, the authors identified several factors, including the E3 ligase substrate adaptor SPOP as regulators of IRF1 protein levels. The screening approach used good controls (such as Myc instead of IRF1) and showed significant results. Further, the authors identified potential SPOP-binding motifs in IRF1 and showed that they (or at least one of them) are important for the recognition and turnover via SPOP; they used flow cytometry, co-immunoprecipitation, western immunoblotting, and dual luciferase assays. Finally, to determine the response of IRF1 transcriptional activity to the presence and absence of SPOP, the authors performed a qPCR analysis of IRF1-driven transcripts, which revealed a significant increase in SPOP knockout cells.The manuscript is well written. The authors find another substrate of SPOP. One open question is how significant this E3 ligase is for the turnover of IRF1. The difference in ubiquitination in WT and SPOP-binding motif-devoid IRF1 is modest. The differences in transcriptional activity, however, are large. A set of additional experiments are necessary to understand these counterintuitive effects. The authors identify 4 SPOP-binding motifs in IRF1 and suggest that they work synergistically in the recognition of IRF1 by SPOP and its subsequent degradation. However, without testing the contribution of individual SPOP-binding motifs, it is impossible to say whether they all contribute to SPOP turnover.1. The current version of Figure 1E is weak in supporting the idea that IRF1 degradation is independent of autophagy/autolysosomal degradation. Additional controls including time-dependent CHX treatment (like in Figure 1A) and -IFNγ in the presence of Bafilomycin will help to answer this question clearly.

New data addressing the contribution of autophagy/lysosomal degradation to IRF1 turn-over in a protein stability chase were added as new figures Figure 4—figure supplement 1A-B. IRF1 stability was investigated in the presence of both cycloheximide and Bafilomycin A. From these data, we concluded that IRF1 is predominantly degraded by the proteasome, with little to no contribution by lysosome-dependent mechanisms.

2. Is IRF1 ubiquitinated by SPOP also in the absence of inflammation or is this an inflammation-specific response?

Exogenously expressed IRF1 is rapidly turned-over in a SPOP-dependent manner in the absence of any cytokine stimulation, indicating that this process does likely not majorly rely on inflammatory signals. Most experiments with exogenously expressed IRF1 presented in the manuscript were performed in the absence of cytokine stimulation, including the genetic screen that identified SPOP.

3. There are no error bars in some histogram plots (Figure 3E, Figure 3F, Figure 4J, Figure 5C). How many biological and technical replicates did the authors perform?

Error bars were added to the respective figures and the number of replicates noted in their figure legends.

4. In Figure 4D, it seems that both IRF1 and SPOP localize in the nucleus but the resolution of the microscopy image is not sufficient to clearly understand their subcellular distribution. SPOP is often localized to nuclear bodies such as nuclear speckles. Micrographs with better resolution and concentration on the nucleus are required.

New data in Figures 4A and Figure 4—figure supplement 1A address the subcellular localization of IRF1 and SPOP by confocal immunofluorescence microscopy in two cell models. From these new data, we conclude that SPOP localizes in both of our cell models diffusely in the nucleoplasm and nuclear speckles, whereas IRF1 is exclusively diffusely nucleoplasmic. Additional data show that IRF1 only accumulates as a diffusely nucleoplasmic pool upon proteasome inhibition. Together, these results indicate that IRF1 and SPOP likely interact in the nucleoplasm, and that IRF1 is predominantly degraded in the nucleoplasm, with limited involvement of SPOP in nuclear speckles. Text discussing possible reasons for the discrepancy with data presented by Gao *et al.* was added to the Discussion section.

5. The decrease in the ubiquitylation level of IRF1 with mutated SB motifs compared to IRF1 WT is modest (Figure 4I). SPOP seems to be responsible for the turnover of only a small fraction of IRF1. The authors found other E3 ligase hits in their genetic screen, e.g. HUWE1 (a strong candidate hit in Figure S2F). Have they considered testing whether there are other E3 ligases involved in the turnover of IRF1?

UBE3C and HUWE1 were two other E3 ligases identified in the IRF1 screen. UBE3C was not further pursued, as the fact that it was also a strong hit in the cMYC screen indicated this may be a more general regulator of proteasomal degradation, consistent with published data (Chu et al. JBC 2013, PMID: 24158444). The role of HUWE1 in IRF1 turn-over is being further elucidated outside of this manuscript’s scope, which may be indirect as we recently showed for the inflammation regulator tristetraprolin (Scinicariello *et al.*, *eLife* 2023).

6. In Figure 5A and 5B, to be able to unambivalently identify whether tested transcripts are only driven by the transcriptional activity of IRF1 but not by other IRF family members or other transcription factors when treated with IFNγ, a control condition like sgIRF1 cells are necessary.

New data presented in Figures 5A/B + Figure 5—figure supplement 1A demonstrate that in *SPOP* knock-out cells (which have increased IRF1), the mRNA levels of IRF1 target genes are increased. Importantly, this new data set includes *SPOP/IRF1* double knock-out epistasis samples, in which this effect was lost. Together, these data show that SPOP is relevant for curtailing IRF1-dependent IFN-stimulated gene expression.

7. Experiments in Figure 5C and Figure 5D show a comparison of +/- WT SPOP expression. In Figure 5C, the ratio is about 10fold, but in Figure 5D around 100 fold. What results in this large deviation?

We acknowledge this difference, but would like to point out that irrespective of the effect magnitude, the qualitative behavior is comparable in both panels. Moreover, the internally controlled nature of the dual luciferase setup, makes us confident that both data sets are reliable. Although we cannot identify a particular cause for the difference with certainty, a higher transfection efficiency and higher signal gain settings on the plate reader for the second experiment (Figure 5D) may have improved the experimental range of the assay, and consequently may have resulted in larger effect differences upon SPOP expression.

8. The authors observe little to no protein turnover effect of SPOP on cMYC. However, it has been previously shown that SPOP recognizes cMYC as a ubiquitination substrate (PMID: 28414305, PMID: 30002443). Could the authors please discuss the potential reasons for this discrepancy?

These references and accompanying text addressing this point were added to the Discussion section of the revised manuscript.

9. The authors mentioned a recent study (Tawaratsumida et al., 2022) that shows that deletion of SPOP results in LYN-dependent (Src kinase family member LYN) degradation of IRF1, and inhibition of its transcriptional activity. The discussion (in lines 452-456) suggests that the differences may arise from the expression of TLRs and associated adapters exclusively in immune cells. However, colon/colorectal cancer cells including RKO cells are known to express most TLRs (e.g. PMID: 33244454). It thus seems that TLR expression is not the critical difference that causes the differing response. Could the authors please discuss what other biological reasons could potentially cause this different response?

This reference, and text discussing this point was added to the Discussion section.

10. The statement "…the four predicted degrons collectively are required for SPOP-dependent IRF1 degradation." (lines 298-299) cannot be made without testing the response of the SPOP degrons individually, as well as in binary and ternary combinations. It is possible that only one of the motifs is responsible for the observed effect.

We agree, and the text in the Abstract and manuscript are adapted to reflect this notion.